# Federated Learning with Efficient Local Adaptation for Realized Volatility Prediction

**Lei Zhao**                                                   *leizhao@uvic.ca*
*Department of Electrical and Computer Engineering*
*University of Victoria*

**Lin Cai**                                                    *cai@ece.uvic.ca*
*Department of Electrical and Computer Engineering*
*University of Victoria*

**Wu-Sheng Lu**                                               *wslu@ece.uvic.ca*
*Department of Electrical and Computer Engineering*
*University of Victoria*

**Reviewed on OpenReview:** *https://openreview.net/forum?id=bHdEtW5E7O*

## Abstract

Financial markets present unique challenges for Federated Learning (FL) due to fragmented datasets, dynamic participation, and the critical need for precise and reliable predictions. Isolated local datasets often fail to capture the full spectrum of market dynamics, blocking accurate realized volatility predictions. Unlike traditional FL methods that focus on improving convergence during the training process, we propose Federated Learning with Adaptive Robustness and Efficiency for Local Adaptation (FLARE-LA), a novel framework designed to optimize predictive performance after the global training phase. FLARE-LA leverages Taylor-based local linearization and probabilistic optimization to efficiently adapt global models to local data distributions, enabling fast responsiveness to new market conditions. This adaptability ensures trained local models align with real-world scenarios, making FLARE-LA particularly suited to dynamic financial applications. Extensive experimental evaluations demonstrate FLARE-LA's superior performance, showcasing its ability to significantly enhance post-FL outcomes compared to state-of-the-art FL algorithms. The results underscore FLARE-LA's unique capability to drive advancements in financial forecasting and other high-stakes, rapidly evolving domains.

## 1 Introduction

Predicting realized volatility is a cornerstone of financial forecasting, essential for effective risk management and informed investment strategies within the framework of deep hedging (Buehler et al., 2019; Vuletić & Cont, 2023; Mueller et al., 2024). However, financial markets naturally generate fragmented and asynchronous data across multiple trading venues. These platforms are unable to share data with third parties due to stringent privacy concerns, regulatory constraints, and technical challenges (Kairouz et al., 2021). The data fragmentation poses significant obstacles to the accuracy and reliability of realized volatility predictions. When data is distributed across multiple platforms, the collective market understanding becomes incomplete, limiting the ability to accurately capture price movements and liquidity dynamics. Volatility prediction, which depends on comprehensive market data, is particularly vulnerable to inaccuracies and biases under these conditions. Furthermore, discrepancies in liquidity levels and pricing for the same asset across exchanges exacerbate these challenges, potentially leading to incorrect volatility estimates when one platform's data fails to reflect broader market trends (Otero, 2002; Madhavan, 2000). Federated Learning (FL) offers a promising solution by enabling collaborative model training across distributed data sources while preserving

data privacy, which addresses key privacy and regulatory concerns by ensuring data remains local to each trading platform (Yang et al., 2019; Yu et al., 2020; Tan et al., 2022; Chen et al., 2023a; Meng et al., 2024).

Financial markets impose uniquely demanding requirements on FL due to the critical need for high precision, reliability, and robustness. Unlike other domains, where minor inaccuracies may be tolerable, small errors in financial forecasting can result in significant financial losses or missed opportunities (Ning et al., 2023). Volatility prediction, in particular, presents a highly challenging task as it requires models that not only identify complex and rapidly evolving trends in market behavior but also provide robust and interpretable risk estimates (Bergeron et al., 2021). The fragmented nature of financial data intensifies these challenges. Trading platforms operate independently, each generating datasets that reflect its unique market conditions, liquidity levels, and trading behaviors. The data decentralization introduces substantial obstacles to achieving consistent, high-quality predictions across platforms.

While data heterogeneity and dynamic participation are common challenges in FL, their impact is amplified in financial markets. The inherent variability in datasets across trading platforms leads to significant discrepancies in local and global data distributions. Additionally, trading platforms frequently enter and exit the training process due to operational constraints, creating a dynamic and unpredictable training environment. Maintaining robustness under such conditions is essential to ensuring consistent model performance across all participants. Beyond these technical challenges, the necessity for interpretable and trustworthy predictions is particularly acute in financial applications. Models must go beyond delivering accurate forecasts, which must also quantify uncertainties effectively, enabling informed decision-making in high-stakes environments. Traditional FL methods often lack the precision, adaptability, and interpretability required for such applications, limiting their practical utility in financial forecasting.

To address these challenges, we introduce Federated Learning with Adaptive Robustness and Efficiency for Local Adaptation (FLARE-LA), a cutting-edge framework designed to address the distinct requirements of financial markets. FLARE-LA utilizes Taylor-based linearization to achieve computationally efficient and accurate local adaptations, effectively aligning the global model with platform-specific datasets. Moreover, it incorporates a probabilistic mechanism that leverages the Jacobian matrix of the global model, facilitating localized optimization and delivering interpretable uncertainty quantification to enhance reliability and support informed decision-making. This integrated approach ensures robust performance in dynamic and fragmented environments while maintaining computational efficiency. By blending global insights with fine-tuned local adjustments, FLARE-LA substantially improves the accuracy of realized volatility predictions, satisfying the stringent precision and reliability demands of financial forecasting.

Although FLARE-LA is rigorously evaluated within the context of financial markets, its underlying principles, such as efficient local adaptation and uncertainty-aware predictions, are broadly applicable to a wide range of domains. The financial market serves as a challenging and representative scenario that underscores the framework's capabilities, providing valuable insights into its potential for other complex and dynamic environments. Experimental evaluations demonstrate that FLARE-LA consistently outperforms state-of-the-art baselines, achieving lower mean loss, Value at Risk (VaR95%), and Conditional Value at Risk (CVaR95%) across diverse platform distributions and participation rates. These results highlight the framework's scalability, adaptability, and robustness, making it well-suited for general federated learning tasks where data heterogeneity, dynamic participation, and computational efficiency are critical.

In the following sections, we provide an overview of related work in Section 2, positioning our contributions within the broader landscape of FL and financial forecasting. Section 3 introduces the fragmented nature of financial markets and formulates the problem addressed by our framework. In Section 4, we detail our proposed approach, highlighting the integration of probabilistic frameworks and efficient local adaptation techniques. Empirical evaluations of our method, showcasing its effectiveness and robustness across various scenarios, are detailed in Section 5. Finally, we summarize our findings, and outline potential directions for future research in Section 6.

## 2    Related Work

In the context of financial markets, predicting realized volatility using order book data is a challenging task due to the decentralized nature of data acquisition (Banabilah et al., 2022). Order books, which capture buy and sell orders for securities, form a dynamic and fragmented data environment, often referred to as "data islands." These characteristics make FL an appealing approach for such environments (Hasbrouck, 2007). FL enables collaborative model training across distributed data sources while preserving data privacy, a critical requirement in financial markets.

Existing FL methods face specific challenges when applied to predicting realized volatility in financial markets, such as high data heterogeneity, rapid changes in data patterns, and the need for timely model updates. Methods like FedProx (Li et al., 2020), which introduces a proximal term to the local objective function to stabilize optimization, mitigate the effects of heterogeneity by reducing the impact of local updates that deviate significantly from the global model. However, while FedProx offers stability, its proximal term may not fully capture the dynamic nature of financial data, and it can slow convergence, a critical drawback in fast-paced trading environments (Arthur et al., 2018; Cantillon & Yin, 2011). To address local-global mis-alignments, SCAFFOLD (Karimireddy et al., 2020) incorporates control variates to correct the drift in local updates, improving alignment with the global model. However, the rapidly evolving financial landscape can still lead to misalignments that adversely affect prediction accuracy (Boukherouaa et al., 2021). While effective in certain scenarios, the introduction of control variates increases computational complexity and communication overhead, posing challenges for deployment in high-frequency trading environments.

Personalized FL methods, such as FedPer (Arivazhagan et al., 2019), decouple shared global parameters from client-specific local parameters to provide personalization. Despite this, the high variability and unpredictability in financial markets require frequent adjustments to personalized models, making the process resource-intensive and limiting scalability in large-scale financial networks. Similarly, LG-FedAvg (Liang et al., 2020), APFL (Deng et al., 2020), and pFedMe (T Dinh et al., 2020) adopt approaches to balance global and local knowledge but face challenges in handling the feature and distributional heterogeneity prevalent in financial markets.

Recent innovations, including Ditto (Li et al., 2021), FedRep (Collins et al., 2021), and SuPerFed (Hahn et al., 2022), have explored various personalization techniques, such as interpolating global and local models or applying proximity regularization. These methods highlight progress in adapting global models to client-specific data. However, they often require fine-tuning or additional computational resources for new clients, reducing scalability in highly dynamic environments like financial trading. Meng et al. (2024) explores techniques to enhance global generalization and local personalization through adaptive aggregation and dual optimization, which aligns with our goal of striking a balance between global insights and local adaptations in heterogeneous FL settings. While their work focuses on representation learning and aggregation strategies, our method introduces a Taylor-based linearization approach combined with a probabilistic framework to achieve more precise and interpretable local adaptation. Tan et al. (2022) provides insights into handling client-specific model updates using personalized layers and meta-learning, offering a solution for improving local performance in non-IID settings. This is relevant to our method in FLARE-LA, which also aims to achieve strong local adaptation but does so through efficient linearized updates and probabilistic adjustments, eliminating the need for additional network layers or meta-learning components. Chen et al. (2023a) focuses on sparse model adaptation to enhance scalability and computational efficiency in personalized FL, a strategy particularly useful in resource-constrained environments. Similarly, Yu et al. (2020) highlights the importance of localized training adjustments to address the limitations of federated aggregation in heterogeneous datasets. Both approaches emphasize the need for effective local training, which resonates with our method's focus on dynamic participation and efficient adaptation. However, our approach extends these ideas by leveraging Jacobian-based linearization and uncertainty quantification, enabling robust local updates tailored to the fragmented and rapidly changing nature of financial data.

Advances in neural network behavior further inspire solutions for FL. Research has revealed that infinitely wide deep neural networks (DNNs) exhibit behaviors similar to their Taylor expansions around initialization (Chizat et al., 2019). Extensions of this analysis to finite-width DNNs demonstrate that their training dynamics resemble linear models (Seleznova & Kutyniok, 2022), while the inductive biases of linearized neural networks

effectively summarize full network functions (Maddox et al., 2021). These findings motivate the development of FL approaches that better address the unique characteristics of local trading platforms, where global models often fail to capture localized data patterns.

To address these challenges, the proposed FLARE-LA framework introduces adaptive local training mechanisms that go beyond traditional FL approaches by focusing on post-training performance optimization. FLARE-LA leverages insights from neural network linearization to enable precise and computationally efficient local adaptations, ensuring that global models are effectively refined to meet the specific needs of individual trading platforms. Unlike existing methods, FLARE-LA is designed to address the dynamic and heterogeneous nature of financial markets by rapidly adapting to new data and evolving conditions, ensuring that local models remain robust and aligned with real-world scenarios. The proposed innovative approach is a transformative solution for achieving high-precision and reliable predictions in decentralized and fragmented financial environments.

## 3 Fragmented Financial Markets

### 3.1 Background

In financial markets, trading occurs across a wide range of exchanges and platforms, resulting in highly fragmented datasets. Each platform independently maintains transaction and order book data, capturing buy and sell orders as well as their execution details. This fragmentation provides an incomplete view of market activity for any given asset, with notable variations in pricing, liquidity, and order depth across platforms (Hasbrouck, 2007).

The order book plays a critical role in market analysis, offering traders insights into short-term trading dynamics. By displaying order imbalances and identifying potential support and resistance levels for a stock, the order book supports informed trading decisions. Heightened market activity and uncertainty are often reflected in increased realized volatility, which arises from frequent directional price movements. Trading data, which records executed transactions, complements the order book by offering valuable insights into market dynamics, such as price trends, trading volumes, and liquidity conditions.

Predicting short-term realized volatility is essential for effective risk management and the development of trading strategies (Chen et al., 2023b). By analyzing order book and trade data over fixed time intervals, traders and institutions can forecast future volatility levels, enabling improved decision-making and enhanced risk mitigation. Realized volatility predictions help market participants manage exposure, optimize portfolio allocations, and design robust trading strategies.

Extracting meaningful insights from order book data is vital for understanding market dynamics and assessing stock values. Key metrics, such as the bid-ask spread, weighted average price, and volume-related indicators, provide a wealth of information about market liquidity and potential volatility. However, the fragmented nature of financial markets poses significant challenges for comprehensive analysis, as data silos limit access to the full scope of market activity.

### 3.2 Problem Formulation

By leveraging diverse data sources, FL facilitates the development of a robust global model that enhances local predictions, preserving both data privacy and confidentiality, which allows trading platforms to benefit from a comprehensive understanding of market dynamics while maintaining compliance with regulatory requirements and addressing privacy concerns.

Consider a distributed dataset consisting of $n$ data sample pairs $\{\mathbf{x}_i, y_i\}_{i=1}^{n}$ across $|E|$ trading platforms. Each data sample pair represents features extracted from order book and trading data, with $\mathbf{x}_i$ denoting the feature vector and $y_i$ representing the corresponding label, which is the volatility. There are 363 features for each sample generated from order book and trading data, capturing essential market dynamics such as bid-ask spreads, price movements, and trading volumes.

We denote the local dataset of the $c$-th trading platform as $P_c$, which contains $n_c$ training samples. The union of all local datasets from each trading platform, $P_1 \cup P_2 \cup \cdots \cup P_{|E|}$, covers the entire dataset, guaranteeing that each sample is assigned to exactly one trading platform's dataset. For trading platform $c$, the labels $\{y_i\}_{i \in P_c}$ represent the volatility levels observed in the corresponding platform's trading data. These volatility labels are used as the ground truth for training the predictive model.

We aim to develop a predictive model, represented by a deep neural network function $f$, which maps an input feature vector $\mathbf{x}$ to an output volatility prediction $y$. The model is trained using the distributed dataset across multiple trading platforms, leveraging the features extracted from order book and trading data to predict future volatility levels accurately. The local objective function for trading platform $c$ is defined as

$$\underset{\boldsymbol{w}}{\textbf{minimize}} \quad \boldsymbol{L_c}(\boldsymbol{w}) = \frac{1}{2} \sum_{i \in P_c} \left( f(\boldsymbol{x_i}, \boldsymbol{w}) - y_i \right)^2 \quad \text{for } c = 1, \cdots, |E|, \tag{1}$$

where $\boldsymbol{w}$ represents the trainable parameters of the model. Meanwhile, the global objective function, aggregating the local objectives across all trading platforms, is given by

$$\underset{\boldsymbol{w}}{\textbf{minimize}} \quad \boldsymbol{L}(\boldsymbol{w}) = \frac{1}{|E|} \sum_{c=1}^{|E|} \boldsymbol{L_c}(\boldsymbol{w}). \tag{2}$$

Realized volatility prediction poses unique challenges due to the fragmented nature of financial data, and the rapidly evolving market conditions. Each platform's dataset captures only a localized perspective of the broader market, leading to non-IID data distributions that complicate the development of a unified model. Furthermore, the predictive task demands a model capable of capturing complex, non-linear interactions between features to provide reliable and interpretable outputs. By leveraging FL, trading platforms can train a global model that integrates diverse data sources, improving predictive accuracy while maintaining data privacy. This collaborative framework enables financial institutions to optimize their trading strategies, enhance risk management, and make informed decisions in volatile market conditions.

## 4 Federated Learning with Adaptive Robustness and Efficiency for Local Adaptation

In this section, we introduce our proposed approach FLARE-LA, a framework designed to address the challenges posed by heterogeneous local datasets and dynamic participation in financial markets. While the global objective function in (2) captures general patterns across all trading platforms, it may fail to fully represent the unique characteristics of each platform's local dataset, potentially resulting in suboptimal performance for individual platforms as outlined in (1). To overcome this limitation, FLARE-LA provides an innovative mechanism for trading platforms to adapt the globally trained model to their specific local data.

The FLARE-LA framework operates through a two-stage process. In the initial stage, trading platforms collaboratively train a global model while ensuring strict data privacy. This is accomplished by aggregating model updates from each platform without transmitting raw data, thereby preserving confidentiality and compliance with privacy regulations. The global model, built from the collective knowledge of all participating platforms, effectively captures shared patterns and structures across the distributed datasets. This stage provides a robust baseline for subsequent local adaptation. Moreover, FLARE-LA is designed to seamlessly integrate advanced federated learning techniques into this collaborative training stage, enhancing flexibility and adaptability to various use cases.

In the second stage, FLARE-LA introduces an innovative local adaptation mechanism that fine-tunes the globally trained model to align with the specific characteristics of each trading platform's dataset. This process leverages Taylor-based linearization and probabilistic frameworks to achieve computational efficiency and precision. By utilizing the Jacobian matrix of the global model, FLARE-LA integrates localized optimization with interpretable uncertainty quantification, enabling platforms to adapt the global model dynamically while maintaining robustness in predictive accuracy. This approach ensures that FLARE-LA excels in addressing the challenges of non-IID data distributions, enhancing model performance in diverse and fragmented environments.

By combining the strengths of FL with tailored local adaptations, FLARE-LA effectively addresses the inherent heterogeneity of financial market datasets and accommodates the dynamic participation of trading platforms. This dual-stage approach ensures that each platform benefits from the collaborative insights of FL while achieving optimal performance for its specific market conditions.

### 4.1 Federated Training equipped with Efficient Local Adaptation for Financial Market Dynamics

The initialization of model weights plays a pivotal role in determining the efficiency and stability of the training process. Arbitrary or poorly chosen initialization methods can block progress, leading to issues such as slow convergence or training stagnation (Xie et al., 2017). To address these challenges, it is essential to ensure that the activation distributions maintain consistent variance as the network deepens, preventing common pitfalls like vanishing or exploding gradients. To achieve this, the initial weights are drawn from a Gaussian distribution with a mean of zero and a standard deviation inversely proportional to the square root of the number of input units feeding into the layer as

$$\boldsymbol{w^0} \sim \mathcal{N}\left(0, \frac{1}{\sqrt{n_{\text{in}}}}\right), \tag{3}$$

where $\boldsymbol{w^0}$ denotes the initial weight vector, and $n_{\text{in}}$ represents the number of input units in the layer. This tailored initialization ensures a balanced variance in the activation distributions across layers, fostering a smoother gradient flow and more stable training dynamics.

Federated training operates in a dynamically evolving environment where the participation of trading platforms fluctuates unpredictably. At each training round, a subset of trading platforms, denoted as $S^t \subseteq |E|$, is selected to participate. This mirrors real-world scenarios where platform availability is influenced by operational constraints, market activity, or other factors. To capture these dynamics, the set $S^t$ is sampled from predefined distributions, including Exponential, Geometric, Gamma, and Chi-square distributions, reflecting a variety of participation patterns.

The dynamic nature of platform participation introduces additional complexity to the federated training process, as the global model must adapt to fluctuating contributions without compromising performance. By incorporating realistic participation patterns into our simulation, we ensure that the training procedure reflects the challenges of real-world financial environments, enhancing the robustness and applicability of our approach.

Once the active participants for round $t$ are determined, the current global model $\boldsymbol{w^t}$ is distributed to the selected trading platforms in $S^t$. Each platform initializes its local model for the training round as

$$\{\boldsymbol{w_{c,0}^t} = \boldsymbol{w^t}\}_{c \in S^t}, \tag{4}$$

where $\boldsymbol{w_{c,0}^t}$ represents the initial local model weights for trading platform $c$ at the onset of round $t$. This initialization ensures that all participating platforms begin the round with identical copies of the global model, fostering a collaborative and unified starting point in the dynamic participation environment.

During local training on trading platform $c$, the model undergoes iterative updates using financial market data. The $k$-th step of this update process is defined as:

$$\boldsymbol{w_{c,k+1}^t} = \boldsymbol{w_{c,k}^t} - \alpha_l \boldsymbol{\nabla L_c}(\boldsymbol{w_{c,k}^t}), \tag{5}$$

where $\alpha_l$ is the local learning rate, tailored to the unique dynamics and characteristics of each platform's dataset. This localized learning process allows each platform to refine the model in alignment with its specific market conditions. The local training procedure continues for $K$ iterations, resulting in a final local model as

$$\boldsymbol{w_{c,K}^t} = \boldsymbol{w^t} - \sum_{k=1}^{K} \alpha_l \boldsymbol{\nabla L_c}(\boldsymbol{w_{c,k}^t}), \tag{6}$$

which integrates the cumulative effects of gradient-based updates, highlighting how each platform adapts the global model to its specific data through weighted gradient descents.

To quantify the divergence between the locally adapted model and the initial global model, we define the model discrepancy for trading platform $c$ after $K$ iterations as

$$\triangle \boldsymbol{w_c^t} = \boldsymbol{w_{c,K}^t} - \boldsymbol{w^t}, \tag{7}$$

measuring the extent to which each platform's local updates diverge from the global model parameters, and reflecting the influence of its unique market data on the learning process.

The aggregated local updates are used to compute the global model for the next iteration as

$$\boldsymbol{w^{t+1}} \leftarrow \boldsymbol{w^t} + \frac{\alpha_g^t}{|S^t|} \sum_{c \in S^t} \triangle \boldsymbol{w_c^t}, \tag{8}$$

where $\alpha_g^t$ is the global learning rate for round $t$, and the contribution of each local model is normalized by the number of participating platforms $|S^t|$. This normalization ensures equitable integration of local updates into the global model, promoting fairness and robustness across platforms. The updated global model $\boldsymbol{w^{t+1}}$ marks the conclusion of the $t$-th round of training and serves as the starting point for the next round of FL.

The above iterative process allows FLARE-LA to adaptively refine the global model by incorporating diverse contributions from participating platforms within the dynamic and heterogeneous environment of financial markets. Essentially, the federated iterations in FLARE-LA are designed to be modular, enabling the seamless integration of any advanced FL solutions. This flexibility enhances the scalability and generalization of the framework, allowing it to adapt to evolving methods and leverage state-of-the-art advancements in FL. By combining tailored local training with equitable aggregation, FLARE-LA effectively addresses the challenges of data heterogeneity and fluctuating participation rates, ensuring robust performance and broad applicability.

The global model $\boldsymbol{w}^*$, obtained after FL training, may not be fully optimized or may exhibit poor local performance due to the diverse nature of local datasets and the dynamic participation. Nonetheless, it serves as the baseline for adaptive local training. To derive the local adaptive training strategy, we consider a given neural network model function $f$. We can approximate $f$ around the trained model parameters $\boldsymbol{w}^*$ using a Taylor expansion

$$f(x; \boldsymbol{w}) \approx f(x; \boldsymbol{w}^*) + J_{\boldsymbol{w}^*}(\boldsymbol{x})^T (\boldsymbol{w} - \boldsymbol{w}^*), \tag{9}$$

where $J_{\boldsymbol{w}^*}(\boldsymbol{x})$ denotes the Jacobian matrix of partial derivatives of $f$ with respect to the model parameters at $\boldsymbol{w}^*$, with dimensions $p \times |P_c|$ and we use $p$ to denote the size of the model parameters. This Jacobian represents the sensitivity of the output with respect to changes in the model parameters near $\boldsymbol{w}^*$.

We formulate the probabilistic model governing the output $y$, given input features $x$ extracted from order book and trading data, and model parameters $\boldsymbol{w}$ as

$$p(y \,|\, x, \boldsymbol{w}) = \mathcal{N}\left(f(\boldsymbol{x}; \boldsymbol{w}), \sigma_c^2\right) = \frac{1}{\sqrt{2\pi\sigma_c^2}} e^{-\frac{(y - f(\boldsymbol{x};\boldsymbol{w}))^2}{2\sigma_c^2}}, \tag{10}$$

where $\sigma_c^2$ represents the variance associated with the Gaussian noise, capturing the inherent uncertainty and noise in the model predictions of volatility. This distribution's mean is specified by the linear approximation obtained from the Taylor expansion of $f$, with a variance $\sigma_c^2$.

For volatility prediction in financial markets using FL, deviations from the baseline global model $\boldsymbol{w}^*$ influence the mean prediction through the Jacobian adjustment, while the Gaussian term $\mathcal{N}(0, \sigma_c^2)$ accounts for the stochastic nature of the predictions. This framework establishes a robust basis for trading platforms to adapt and retrain the global model locally, ensuring performance optimization tailored to the unique characteristics of individual datasets.

For each trading platform $c$ with its local dataset $\{(\boldsymbol{x}_i, y_i)\}_{i=1}^{|P_c|}$, the likelihood function quantifies the probability of observing the given data. It incorporates both the individual variances from the Gaussian noise and the deviations of the model predictions from actual data points. This integration is captured by the model's output and its linear approximation around $\boldsymbol{w}^*$ which is formulated as

$$P_c(\boldsymbol{w}) = \frac{1}{(2\pi\sigma_c^2)^{\frac{|P_c|}{2}}} \exp\left(-\frac{1}{2\sigma_c^2} \sum_{i=1}^{|P_c|} (y_i - (f(\boldsymbol{x}_i; \boldsymbol{w}^*) + J_{\boldsymbol{w}^*}(\boldsymbol{x}_i)^T (\boldsymbol{w} - \boldsymbol{w}^*)))^2\right), \tag{11}$$

which enables trading platforms to effectively assess the fit between their local data and the global model, guiding them in refining the model parameters to better capture the underlying patterns in volatility dynamics.

For rapid local adaptation within our financial market volatility prediction, we transform the likelihood function into its logarithmic form as

$$\log(P_c(\boldsymbol{w})) = -\frac{|P_c|}{2} \log(2\pi\sigma_c^2) - \frac{1}{2\sigma_c^2} \sum_{i=1}^{|P_c|} (y_i - (f(\boldsymbol{x}_i; \boldsymbol{w}^*) + J_{\boldsymbol{w}^*}(\boldsymbol{x}_i)^T(\boldsymbol{w} - \boldsymbol{w}^*)))^2, \tag{12}$$

which simplifies the expression by converting the product of probabilities into a sum of logarithms, linearizing the effects of the parameters and enhancing the tractability of the optimization problem. Importantly, $-\frac{1}{2\sigma_c^2} \sum_{i=1}^{|P_c|} (y_i - f(\boldsymbol{x}_i; \boldsymbol{w}))^2$ represents the sum of squared residuals, adjusted by the inverse of the noise variance $\sigma_c^2$. Therefore, the local adaptation process can be formulated as minimizing the following loss function

$$\hat{L}_c(\boldsymbol{w}) = \frac{1}{2\sigma_c^2} \sum_{i=1}^{|P_c|} (y_i - (f(\boldsymbol{x}_i; \boldsymbol{w}^*) + J_{\boldsymbol{w}^*}(\boldsymbol{x}_i)^T(\boldsymbol{w} - \boldsymbol{w}^*)))^2 + \frac{|P_c|}{2} \log(2\pi\sigma_c^2), \tag{13}$$

which comprises a term that evaluates the sum of squared deviations between the predicted volatility and the actual volatility, scaled by the noise variance $\sigma_c^2$, and a constant term that standardizes the loss based on the dataset size and noise level in the context of local financial market data.

We define $\boldsymbol{J}_{\boldsymbol{w}^*} = \{\boldsymbol{J}_{\boldsymbol{w}^*}(\boldsymbol{x}_i)\}_{i=1}^{P_c}$ as the collection of Jacobian matrices of the model's predictions with respect to the features generated from order book and trading data, evaluated at $\boldsymbol{w}^*$. The sum of the outer products of these Jacobian matrices across all data points forms a symmetric matrix as

$$\sum_{i=1}^{|P_c|} \boldsymbol{J}_{\boldsymbol{w}^*}(\boldsymbol{x}_i) \boldsymbol{J}_{\boldsymbol{w}^*}(\boldsymbol{x}_i)^T = \boldsymbol{J}_{\boldsymbol{w}^*} \boldsymbol{J}_{\boldsymbol{w}^*}^T, \tag{14}$$

which reflects the covariance structure of the gradients, capturing the sensitivity of the model's predictions to the features derived from the trading platforms' data. To simplify computations in practice, the loss function can be reformulated as

$$\begin{aligned} \hat{L}_c(\boldsymbol{w}) = {} & (\boldsymbol{w} - \boldsymbol{w}^*)^T \frac{1}{2\sigma_c^2} \boldsymbol{J}_{\boldsymbol{w}^*} \boldsymbol{J}_{\boldsymbol{w}^*}^T (\boldsymbol{w} - \boldsymbol{w}^*) - (\boldsymbol{w} - \boldsymbol{w}^*)^T \frac{1}{\sigma_c^2} \boldsymbol{J}_{\boldsymbol{w}^*} (\boldsymbol{y_c} - \boldsymbol{f_c}) \\ & + \frac{1}{2\sigma_c^2} (\boldsymbol{y_c} - \boldsymbol{f_c})^T (\boldsymbol{y_c} - \boldsymbol{f_c}) + \frac{|P_c|}{2} \log(2\pi\sigma_c^2), \end{aligned} \tag{15}$$

where $\boldsymbol{f_c} = \{f(\boldsymbol{x}_i; \boldsymbol{w}^*)\}_{i=1}^{P_c}$ and $\boldsymbol{y_c} = \{y_i\}_{i=1}^{P_c}$. It quantifies the balance between the model's internal predictions and the observed deviations from the actual volatility outcomes, scaled by the noise variance, $\sigma_c^2$. This local loss function is critical for adapting the global model to better fit the specific characteristics of the local trading platform's data. The local model adaptation is achieved by setting the gradient of the designed local loss function, $\nabla \hat{L}_c(\boldsymbol{w})$, to zero as

$$\nabla \hat{L}_c(\boldsymbol{w}) = \frac{1}{\sigma_c^2} \boldsymbol{J}_{\boldsymbol{w}^*} \boldsymbol{J}_{\boldsymbol{w}^*}^T (\boldsymbol{w} - \boldsymbol{w}^*) - \frac{1}{\sigma_c^2} \boldsymbol{J}_{\boldsymbol{w}^*} (\boldsymbol{y_c} - \boldsymbol{f_c}) = \boldsymbol{0}. \tag{16}$$

By solving the stationary condition of the linearized local loss, we identify the local adaptation for platform $c$ as

$$\boldsymbol{w} = (\boldsymbol{J}_{\boldsymbol{w}^*} \boldsymbol{J}_{\boldsymbol{w}^*}^T)^{-1} \boldsymbol{J}_{\boldsymbol{w}^*} (\boldsymbol{y_c} - \boldsymbol{f_c}) + \boldsymbol{w}^*, \tag{17}$$

which typically serves as a minimum for a well-defined convex function. This formulation indicates that the local model adaptation is proportional to the inverse of the aggregated Jacobian product, adjusted by the residuals between the observed volatility $\boldsymbol{y_c}$ and the model's baseline predictions $\boldsymbol{f_c}$. Importantly, the matrix-vector product $(\boldsymbol{J}_{\boldsymbol{w}^*} \boldsymbol{J}_{\boldsymbol{w}^*}^T)^{-1} \boldsymbol{J}_{\boldsymbol{w}^*}$ only needs to be computed once per platform, providing significant computational efficiency during repeated local adaptations. And

$$\hat{y}_i = f(\boldsymbol{x}_i; \boldsymbol{w}^*) + J_{\boldsymbol{w}^*}(\boldsymbol{x}_i)^T (\boldsymbol{J}_{\boldsymbol{w}^*} \boldsymbol{J}_{\boldsymbol{w}^*}^T)^{-1} \boldsymbol{J}_{\boldsymbol{w}^*} (\boldsymbol{y_c} - \boldsymbol{f_c}) + \mathcal{N}(0, \sigma_c^2), \tag{18}$$

governs the local model's predictions for a new data sample $\boldsymbol{x}_i$, where the term $\mathcal{N}(0, \sigma_c^2)$ captures the inherent noise variance in volatility forecasting.

Although (17) and (18) assume $\boldsymbol{J}_{\boldsymbol{w}^*} \boldsymbol{J}_{\boldsymbol{w}^*}^{\boldsymbol{T}}$ is invertible, rank-deficient or ill-conditioned Jacobians can arise under real-world data constraints, e.g., when local data are limited or collinear. To ensure reliable adaptation in such scenarios, we introduce a small regularization parameter $\lambda > 0$ on the diagonal of $\boldsymbol{J}_{\boldsymbol{w}^*} \boldsymbol{J}_{\boldsymbol{w}^*}^{\boldsymbol{T}}$, and we can obtain

$$\boldsymbol{w} = \boldsymbol{w}^* + \left( \boldsymbol{J}_{\boldsymbol{w}^*} \boldsymbol{J}_{\boldsymbol{w}^*}^{\boldsymbol{T}} + \lambda \boldsymbol{I} \right)^{-1} \boldsymbol{J}_{\boldsymbol{w}^*} \left( \boldsymbol{y_c} - \boldsymbol{f_c} \right). \tag{19}$$

In this modification, every eigenvalue of $\boldsymbol{J}_{\boldsymbol{w}^*} \boldsymbol{J}_{\boldsymbol{w}^*}^{\boldsymbol{T}}$ is shifted by at least $\lambda$, thereby ensuring positive definiteness and invertibility. Therefore, the predictive step becomes

$$\hat{y}_i = f(\boldsymbol{x}_i; \boldsymbol{w}^*) + J_{\boldsymbol{w}^*}(\boldsymbol{x}_i)^T \left( \boldsymbol{J}_{\boldsymbol{w}^*} \boldsymbol{J}_{\boldsymbol{w}^*}^{\boldsymbol{T}} + \lambda \boldsymbol{I} \right)^{-1} \boldsymbol{J}_{\boldsymbol{w}^*} (\boldsymbol{y_c} - \boldsymbol{f_c}) + \mathcal{N}(0, \sigma_c^2). \tag{20}$$

This diagonal shift has minimal impact on the model's geometry while resolving degenerate eigenvalues and stabilizing local adaptation. Therefore, FLARE-LA accommodates a wide range of data conditions and ensures the final adaptation step remains both computationally feasible and theoretically sound, even when the Jacobian product matrix is poorly conditioned.

---

**Algorithm 1** Federated Learning with Adaptive Robustness and Efficiency for Local Adaptation (FLARE-LA)

---

**Input:** Global model initialization $\boldsymbol{w}^0$, learning rates $\alpha_l, \{\alpha_g^t\}$, participation distribution, local datasets $\{\mathcal{D}_c\}_{c \in \mathcal{C}}$, regularization parameter $\lambda \geq 0$, ill-condition tolerance $\varepsilon > 0$.
**Output:** Updated global model $\boldsymbol{w}^T$, optionally adapted local models $\{\boldsymbol{w}_c^*\}$

1: **Initialize:** $\boldsymbol{w}^0 \sim \mathcal{N}(0, 1/\sqrt{n_{\mathrm{in}}})$
2: **for** $t = 1$ **to** $T$ **do**
3:    Sample active platforms $S^t \sim$ Participation Distribution
4:    **for each** platform $c \in S^t$ **in parallel do**
5:       Initialize local model $\boldsymbol{w}_{c,0}^t = \boldsymbol{w}^t$
6:       **for** $k = 1$ **to** $K$ **do**
7:          $\boldsymbol{w}_{c,k}^t \leftarrow \boldsymbol{w}_{c,k-1}^t - \alpha_l \nabla L_c(\boldsymbol{w}_{c,k-1}^t)$
8:       **end for**
9:       $\boldsymbol{w}_{c,K}^t \leftarrow$ final local model after $K$ steps
10:       $\triangle \boldsymbol{w}_c^t \leftarrow \boldsymbol{w}_{c,K}^t - \boldsymbol{w}^t$
11:    **end for**
12:    Aggregate global model: $\boldsymbol{w}^{t+1} \leftarrow \boldsymbol{w}^t + \frac{\alpha_g^t}{|S^t|} \sum_{c \in S^t} \triangle \boldsymbol{w}_c^t$
13: **end for**
14: **Set** $\boldsymbol{w}^* \leftarrow \boldsymbol{w}^T$ {Final global model after $T$ rounds}
15: **Efficient Local Adaptation:**
16: **for each** platform $c$ (if local adaptation is needed) **do**
17:    Compute Jacobian $\boldsymbol{J}_{\boldsymbol{w}^*}$ on $\mathcal{D}_c$, and residuals $\boldsymbol{r}_c \leftarrow \boldsymbol{y}_c - \boldsymbol{f}_c$
18:    Compute the eigendecomposition: $\boldsymbol{J}_{\boldsymbol{w}^*} \boldsymbol{J}_{\boldsymbol{w}^*}^T = \mathbf{U} \, \boldsymbol{\Lambda} \, \mathbf{U}^T$
19:    Obtain the smallest eigenvalue: $\lambda_{\min} = \min_i \lambda_i$
20:    **if** $\lambda_{\min} \leq \varepsilon$ **then**
21:       $\boldsymbol{J}_{\boldsymbol{w}^*} \boldsymbol{J}_{\boldsymbol{w}^*}^T \leftarrow \boldsymbol{J}_{\boldsymbol{w}^*} \boldsymbol{J}_{\boldsymbol{w}^*}^T + \lambda \mathbf{I}$
22:    **end if**
23:    Update local model: $\boldsymbol{w}_c^* \leftarrow \boldsymbol{w}^* + (\boldsymbol{J}_{\boldsymbol{w}^*} \boldsymbol{J}_{\boldsymbol{w}^*}^T)^{-1} \boldsymbol{J}_{\boldsymbol{w}^*} \boldsymbol{r}_c$
24:    **Prediction for new data** $\boldsymbol{x}_i$: $\hat{y}_i \leftarrow f(\boldsymbol{x}_i; \boldsymbol{w}^*) + J_{\boldsymbol{w}^*}(\boldsymbol{x}_i)^T (\boldsymbol{J}_{\boldsymbol{w}^*} \boldsymbol{J}_{\boldsymbol{w}^*}^T)^{-1} \boldsymbol{J}_{\boldsymbol{w}^*} \boldsymbol{r}_c + \mathcal{N}(0, \sigma_c^2)$
25: **end for**

---

By combining the baseline prediction using the global model parameters $f(\boldsymbol{x}_i; \boldsymbol{w}^*)$ with a local refinement term $J_{\boldsymbol{w}^*}(\boldsymbol{x}_i)^T (\boldsymbol{J}_{\boldsymbol{w}^*} \boldsymbol{J}_{\boldsymbol{w}^*}^T)^{-1} \boldsymbol{J}_{\boldsymbol{w}^*} (\boldsymbol{y}_c - \boldsymbol{f}_c)$ and explicitly modeling the inherent variability through a Gaussian noise term $\mathcal{N}(0, \sigma_c^2)$, FLARE-LA offers an adaptive strategy tailored to each trading platform's data heterogeneity. This hybrid prediction process leverages the global model as a baseline while swiftly capturing local requirements, ensuring robust and accurate volatility forecasts under dynamic market conditions. When

$J_{w^*} J_{w^*}^T$ is rank-deficient or ill-conditioned, a small diagonal shift $\lambda \mathbf{I}$ is introduced to preserve positive definiteness and maintain a well-defined local update. The full procedure of FLARE-LA, incorporating both federated training and efficient local adaptation, is summarized in Algorithm 1.

## 4.2 Analysis of the Local Adaptation Mechanism

FLARE-LA integrates the federated training phase with a Taylor-based local adaptation mechanism, effectively addressing the challenges of non-IID data and dynamic participation in financial markets. By uniting collective insights from distributed platforms with precise local refinements, FLARE-LA produces a volatility prediction model that balances global consistency with platform-specific accuracy. To complement the federated training phase, FLARE-LA introduces an advanced Taylor-based linearization strategy for computationally efficient and precise local adaptations. By leveraging the Jacobian matrix of the global model, FLARE-LA approximates complex local adjustments, enabling trading platforms to quickly tailor the global model to their unique data distributions without extensive computational overhead. Additionally, the framework integrates probabilistic modeling to capture prediction uncertainties, enhancing interpretability and reliability, which is key for high-stakes financial decision-making. This modular and extensible approach ensures that FLARE-LA remains a scalable, adaptive, and generalizable framework for FL in financial markets and beyond.

FLARE-LA incorporates adaptive regularization and probabilistic modeling to address the volatility and heterogeneity typical in financial data. Each platform is characterized by its own $\sigma_i^2$, encapsulating local noise levels. The Hessian of the local objective, $\nabla^2 \hat{L}_i(\boldsymbol{w}^*)$, is assumed to be bounded within $(\mu_i, \beta_i)$, and we have

$$\mu_i \leq \left\| \frac{1}{\sigma_i^2} \boldsymbol{J}_{\boldsymbol{w}^*} \boldsymbol{J}_{\boldsymbol{w}^*}^T \right\| \leq \beta_i. \tag{21}$$

To ensure numerical stability and predictable convergence during local adaptation, we first provide the analysis that the local gradient w.r.t $\boldsymbol{w}^*$ can be bounded. According to the quadratic upper bound and the linear lower bound of the local objective function, we can obtain the inequality as

$$
\begin{aligned}
L_i(\boldsymbol{w}^*) - L_i(\boldsymbol{w}) &= L_i(\boldsymbol{w}^*) - L_i(\boldsymbol{z}) + L_i(\boldsymbol{z}) - L_i(\boldsymbol{w}) \\
&\leq \boldsymbol{\nabla} L_i(\boldsymbol{w}^*)^T(\boldsymbol{w}^* - \boldsymbol{z}) + \boldsymbol{\nabla} L_i(\boldsymbol{w})^T(\boldsymbol{z} - \boldsymbol{w}) + \frac{\beta_i}{2}\|\boldsymbol{z} - \boldsymbol{w}\|^2 \\
&= \boldsymbol{\nabla} L_i(\boldsymbol{w}^*)^T(\boldsymbol{w}^* - \boldsymbol{w}) + (\boldsymbol{\nabla} L_i(\boldsymbol{w}^*) - \boldsymbol{\nabla} L_i(\boldsymbol{w}))^T(\boldsymbol{w} - \boldsymbol{z}) + \frac{\beta_i}{2}\|\boldsymbol{z} - \boldsymbol{w}\|^2.
\end{aligned}
\tag{22}
$$

We define

$$\boldsymbol{z} = \boldsymbol{w} - \frac{1}{\beta_i}(\boldsymbol{\nabla} L_i(\boldsymbol{w}) - \boldsymbol{\nabla} L_i(\boldsymbol{w}^*)), \tag{23}$$

and then, we have

$$
\begin{aligned}
(\boldsymbol{\nabla} L_i(\boldsymbol{w}^*) - \boldsymbol{\nabla} L_i(\boldsymbol{w}))^T(\boldsymbol{w} - \boldsymbol{z}) &= -\frac{1}{\beta_i}\|\boldsymbol{\nabla} L_i(\boldsymbol{w}^*) - \boldsymbol{\nabla} L_i(\boldsymbol{w})\|^2, \\
\frac{\beta_i}{2}\|\boldsymbol{z} - \boldsymbol{w}\|^2 &= \frac{1}{2\beta_i}\|\boldsymbol{\nabla} L_i(\boldsymbol{w}^*) - \boldsymbol{\nabla} L_i(\boldsymbol{w})\|^2,
\end{aligned}
\tag{24}
$$

hence,

$$L_i(\boldsymbol{w}^*) - L_i(\boldsymbol{w}) \leq \boldsymbol{\nabla} L_i(\boldsymbol{w}^*)^T(\boldsymbol{w}^* - \boldsymbol{w}) - \frac{1}{2\beta_i}\|\boldsymbol{\nabla} L_i(\boldsymbol{w}^*) - \boldsymbol{\nabla} L_i(\boldsymbol{w})\|^2, \tag{25}$$

which leads to

$$L_i(\boldsymbol{w}) - L_i(\boldsymbol{w}^*) - \boldsymbol{\nabla} L_i(\boldsymbol{w}^*)^T(\boldsymbol{w} - \boldsymbol{w}^*) \geq \frac{1}{2\beta_i}\|\boldsymbol{\nabla} L_i(\boldsymbol{w}^*) - \boldsymbol{\nabla} L_i(\boldsymbol{w})\|^2. \tag{26}$$

Since

$$\frac{1}{|E|}\sum_{i \in E}(L_i(\boldsymbol{w}) - L_i(\boldsymbol{w}^*)) = L(\boldsymbol{w}) - L^*, \tag{27}$$

then, we have

$$2\beta_i(L(\boldsymbol{w}) - L^*) \geq \frac{1}{|E|} \sum_{i \in E} \|\boldsymbol{\nabla} \boldsymbol{L_i}(\boldsymbol{w}) - \boldsymbol{\nabla} \boldsymbol{L_i}(\boldsymbol{w^*})\|^2. \tag{28}$$

The bound on the local gradient can be found as

$$\begin{aligned}
\frac{1}{|E|} \sum_{i \in E} \|\boldsymbol{\nabla} \boldsymbol{L_i}(\boldsymbol{w})\|^2 &= \frac{1}{|E|} \sum_{i \in E} \|\boldsymbol{\nabla} \boldsymbol{L_i}(\boldsymbol{w}) - \boldsymbol{\nabla} \boldsymbol{L_i}(\boldsymbol{w^*}) + \boldsymbol{\nabla} \boldsymbol{L_i}(\boldsymbol{w^*})\|^2 \\
&\leq \frac{2}{|E|} \sum_{i \in E} \|\boldsymbol{\nabla} \boldsymbol{L_i}(\boldsymbol{w}) - \boldsymbol{\nabla} \boldsymbol{L_i}(\boldsymbol{w^*})\|^2 + \frac{2}{|E|} \sum_{i \in E} \|\boldsymbol{\nabla} \boldsymbol{L_i}(\boldsymbol{w^*})\|^2 \\
&\leq 4\beta_i(L(\boldsymbol{w}) - L^*) + \frac{2}{|E|} \sum_{i \in E} \|\boldsymbol{\nabla} \boldsymbol{L_i}(\boldsymbol{w^*})\|^2.
\end{aligned} \tag{29}$$

Therefore, we define the local gradient w.r.t $\boldsymbol{w^*}$ is bounded by $\epsilon_i$, i.e., $\|\boldsymbol{\nabla} \hat{\boldsymbol{L}}_i(\boldsymbol{w^*})\| \leq \epsilon_i$, and $\boldsymbol{w_i^*}$ is the optimal model for trading platform $i$. The improvement of local adaptation in the model space can be bounded by

$$\begin{aligned}
\|\boldsymbol{w} - \boldsymbol{w_i^*}\| &= \|\boldsymbol{w^*} - (\boldsymbol{\nabla^2} \hat{\boldsymbol{L}}_i(\boldsymbol{w^*}))^{-1} \boldsymbol{\nabla} \hat{\boldsymbol{L}}_i(\boldsymbol{w^*}) - \boldsymbol{w_i^*}\| \\
&= \|(\boldsymbol{\nabla^2} \hat{\boldsymbol{L}}_i(\boldsymbol{w^*}))^{-1}[\boldsymbol{\nabla} \hat{\boldsymbol{L}}_i(\boldsymbol{w^*}) + \boldsymbol{\nabla^2} \hat{\boldsymbol{L}}_i(\boldsymbol{w^*})(\boldsymbol{w_i^*} - \boldsymbol{w^*})]\|.
\end{aligned} \tag{30}$$

Since

$$\boldsymbol{\nabla} \hat{\boldsymbol{L}}_i(\boldsymbol{w_i^*}) = \boldsymbol{\nabla} \hat{\boldsymbol{L}}_i(\boldsymbol{w^*}) + \boldsymbol{\nabla^2} \hat{\boldsymbol{L}}_i(\boldsymbol{w^*})(\boldsymbol{w_i^*} - \boldsymbol{w^*}), \tag{31}$$

we obtain that

$$\|\boldsymbol{w} - \boldsymbol{w_i^*}\| = \|(\boldsymbol{\nabla^2} \hat{\boldsymbol{L}}_i(\boldsymbol{w^*}))^{-1}[\boldsymbol{\nabla} \hat{\boldsymbol{L}}_i(\boldsymbol{w_i^*}) - \boldsymbol{\nabla} \hat{\boldsymbol{L}}_i(\boldsymbol{w^*})] + (\boldsymbol{\nabla^2} \hat{\boldsymbol{L}}_i(\boldsymbol{w^*}))^{-1} \boldsymbol{\nabla} \hat{\boldsymbol{L}}_i(\boldsymbol{w^*})\|. \tag{32}$$

We assume the local gradient w.r.t $\boldsymbol{w^*}$ is bounded by $\epsilon_i$. Then, we have

$$\|(\boldsymbol{\nabla^2} \hat{\boldsymbol{L}}_i(\boldsymbol{w^*}))^{-1} \boldsymbol{\nabla} \hat{\boldsymbol{L}}_i(\boldsymbol{w^*})\| \leq \|(\boldsymbol{\nabla^2} \hat{\boldsymbol{L}}_i(\boldsymbol{w^*}))^{-1}\| \|\boldsymbol{\nabla} \hat{\boldsymbol{L}}_i(\boldsymbol{w^*})\| \leq \frac{\epsilon_i}{\mu_i}. \tag{33}$$

Furthermore, we have

$$\begin{aligned}
\|(\boldsymbol{\nabla^2} \hat{\boldsymbol{L}}_i(\boldsymbol{w^*}))^{-1}[\boldsymbol{\nabla} \hat{\boldsymbol{L}}_i(\boldsymbol{w_i^*}) - \boldsymbol{\nabla} \hat{\boldsymbol{L}}_i(\boldsymbol{w^*})]\| &\leq \|(\boldsymbol{\nabla^2} \hat{\boldsymbol{L}}_i(\boldsymbol{w^*}))^{-1}\| \|\boldsymbol{\nabla} \hat{\boldsymbol{L}}_i(\boldsymbol{w_i^*}) - \boldsymbol{\nabla} \hat{\boldsymbol{L}}_i(\boldsymbol{w^*})\| \\
&\leq \frac{\beta_i}{\mu_i} \|\boldsymbol{w_i^*} - \boldsymbol{w^*}\|.
\end{aligned} \tag{34}$$

Therefore, we have

$$\|\boldsymbol{w} - \boldsymbol{w_i^*}\| \leq \frac{\beta_i}{\mu_i} \|\boldsymbol{w_i^*} - \boldsymbol{w^*}\| + \frac{\epsilon_i}{\mu_i} \tag{35}$$

to illustrate how closely each platform's adapted weights, $\boldsymbol{w}$, align with its own local optimum, $\boldsymbol{w_i^*}$. This result shows that the deviation of the adapted solution from the local optimum depends on the curvature factors $(\mu_i, \beta_i)$ and the magnitude of the global model's misalignment with the true local optimum. Therefore, even if the global model $\boldsymbol{w^*}$ is not fully optimized for platform $i$, FLARE-LA's closed-form local adaptation draws the parameters significantly closer to $\boldsymbol{w_i^*}$.

Another key benefit of local adaptation appears in the reduction of the local objective function. By expanding $\hat{L}_i(\boldsymbol{w})$ in a second-order Taylor series around $\boldsymbol{w^*}$, it can be shown that

$$\hat{L}_i(\boldsymbol{w}) = \hat{L}_i(\boldsymbol{w^*}) - \frac{1}{2} \nabla \hat{L}_i(\boldsymbol{w^*})^T (\nabla^2 \hat{L}_i(\boldsymbol{w^*}))^{-1} \nabla \hat{L}_i(\boldsymbol{w^*}). \tag{36}$$

Hence, the local adaptation step decreases the objective by

$$\hat{L}_i(\boldsymbol{w^*}) - \hat{L}_i(\boldsymbol{w}) = \frac{1}{2} \nabla \hat{L}_i(\boldsymbol{w^*})^T (\nabla^2 \hat{L}_i(\boldsymbol{w^*}))^{-1} \nabla \hat{L}_i(\boldsymbol{w^*}). \tag{37}$$

Since $\nabla \hat{L}_i(\boldsymbol{w}^*)$ is proportional to $\frac{1}{\sigma_i^2} \boldsymbol{J}_{\boldsymbol{w}^*}(\boldsymbol{y_i} - \boldsymbol{f_i})$, the extent of this loss reduction is bounded by curvature terms and the squared residual $\|\boldsymbol{y_i} - \boldsymbol{f_i}\|^2$, giving

$$\hat{L}_i(\boldsymbol{w}^*) - \hat{L}_i(\boldsymbol{w}) \leq \frac{\beta_i}{\sigma_i^2 \, \mu_i} \|\boldsymbol{y_i} - \boldsymbol{f_i}\|^2. \tag{38}$$

This result confirms that local adaptation can substantially cut down on the local error to provide the promising local adaptation.

By combining global insights with localized precision, FLARE-LA addresses the fragmentation of market data without sacrificing data privacy or model stability. Its Taylor-based linearization efficiently refines model parameters, while the adaptive regularization and variance-aware modeling capture uncertainty in volatile markets. The bounding of Hessian eigenvalues and gradients ensures that each trading platform's updates remain well-controlled and that performance gains are both significant and predictable. Through the seamless integration of federated collaboration and an efficient yet robust local adaptation mechanism, FLARE-LA ensures enhanced performance across diverse and dynamic environments.

## 5 Experimental Evaluation

This experimental evaluation validates the efficacy and adaptability of the proposed FLARE-LA framework in addressing the challenges of FL across both domain-specific and general scenarios. Our primary focus lies in the financial domain, where the demands for high precision, robustness, and scalability are particularly pronounced. We first utilize a dataset for realized volatility prediction, consisting of order book and trade data from multiple trading platforms. These experiments aim to demonstrate FLARE-LA's ability to handle extreme data heterogeneity, dynamic participation, and the fragmented nature of financial datasets while maintaining robust predictive performance.

To further evaluate the generalizability of FLARE-LA, we extend our experiments to CIFAR10 and MNIST, two well-established datasets in FL research. These datasets allow us to test FLARE-LA's performance under non-IID data distributions, varying client participation rates, and label noise scenarios, mimicking real-world challenges. By incorporating these datasets, we provide complementary evidence of FLARE-LA's versatility and scalability, demonstrating its utility across diverse applications beyond financial forecasting.

In our experiments, client participation is dynamically regulated using a participation ratio, simulating high variability in client engagement during federated training. Non-IID data distributions are modeled using a Dirichlet distribution, with the concentration parameter $\alpha$ controlling the heterogeneity of client data. For $\alpha \to 0$, clients primarily have data from a single class, while $\alpha \to \infty$ results in a uniform distribution of classes across clients. We evaluate the performance of FLARE-LA against several state-of-the-art FL methods, including FedProx (Li et al., 2020), SCAFFOLD (Karimireddy et al., 2020), FedPer (Arivazhagan et al., 2019), LG-FedAvg (Liang et al., 2020), pFedMe (T Dinh et al., 2020), Ditto (Li et al., 2021), FedRep (Collins et al., 2021), and SuPerFed (Hahn et al., 2022). For local training, we utilize the ResNet model, which provides a robust architecture for handling diverse data distributions. The evaluation metrics include mean loss, VaR95%, and CVaR95%, offering a comprehensive assessment of model performance. These metrics underscore FLARE-LA's ability to deliver superior predictive accuracy, adapt effectively to dynamic environments, and maintain computational efficiency, even in challenging FL scenarios.

### 5.1 Experiments on Realized Volatility Prediction

We aim to forecast short-term volatility for stocks spanning multiple sectors (Andrew Meyer, 2021). The dataset comprises both order book and trade data for these stocks, aggregated into multiple time buckets. The values in the order book represent the latest snapshots of market activity, taken at one-second intervals. Each time bucket comprises order book data spanning the 600 seconds. Our experiments involve predicting the volatility for each time bucket of the stocks. There are $428,932$ samples in the entire dataset, where 107 of the stocks have data for 3830 time buckets, while 3 stocks have data for 3829 time buckets, 1 stock has data for 3820 time buckets, and another stock has data for 3815 time buckets. The entire dataset is divided into $10,000$ trading platforms based on a Dirichlet distribution-based non-IID setting (Hsu et al.,

2019). The Dirichlet distribution's concentration parameter, $\alpha$, determines the stock distribution for each trading platform which is set to 0.5 in our experiments. Each trading platform randomly splits its data into a training set and a test set, with 20% allocated for testing. This setup allows us to estimate the performance of each FL algorithm on each trading platform's test set using its personalized model.

### 5.1.1 Performance Comparison

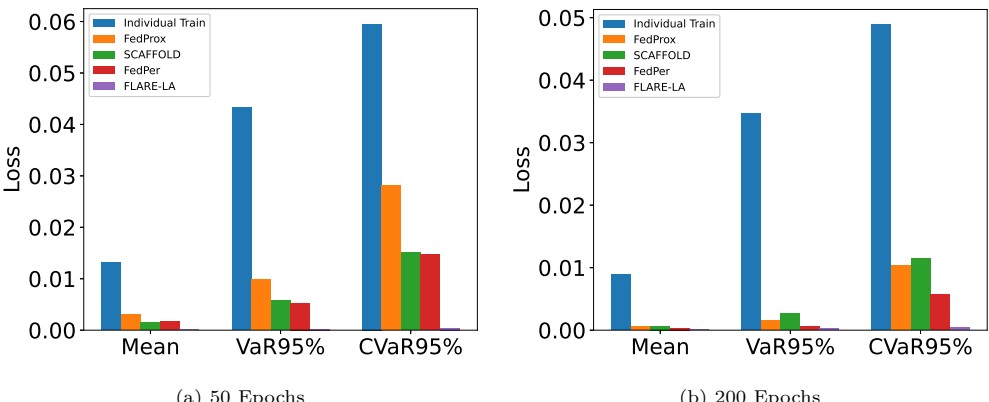

(a) 50 Epochs        (b) 200 Epochs

Figure 1: Comparison of FLARE-LA with Individual Train and other FL baselines (FedProx, SCAFFOLD, and FedPer) across different epochs.

In Fig. 1, we compare the performance of FLARE-LA against Individual Train and baseline FL methods as FedProx, SCAFFOLD, and FedPer over 50 epochs as shown in Fig. 1(a) and 200 epochs as shown in Fig. 1(b). The Individual Train baseline performs training independently for each trading platform without leveraging federated collaboration. Despite identical numbers of parameter updates, FLARE-LA demonstrates significantly superior performance, emphasizing the value of FL in leveraging global insights while tailoring models to local data.

As shown in Fig. 1(a), after 50 epochs, FLARE-LA achieves a remarkably lower mean loss of $7.726 \times 10^{-5}$ compared to Individual Train (0.0132), FedProx (0.0031), SCAFFOLD (0.0015), and FedPer (0.0017). In Fig. 1(b), after 200 epochs, FLARE-LA continues to outperform all baselines, maintaining its lead in terms of mean loss, VaR95%, and CVaR95%. The results highlight FLARE-LA's ability to balance global knowledge with precise local adaptation, resulting in superior performance in federated settings. This demonstrates that FLARE-LA not only accelerates convergence but also ensures higher accuracy and robustness compared to individual and baseline federated training methods.

Across all experimental settings as shown in Fig. 2, FLARE-LA consistently outperforms baseline methods, including FedProx, SCAFFOLD, FedPer, and SuPerFed, in terms of Mean Loss, VaR95%, and CVaR95% for realized volatility prediction tasks.

With low client participation rates as 10%, FLARE-LA demonstrates exceptional robustness and precision. As shown in Fig. 2(a), after 5 training rounds, FLARE-LA achieves a Mean Loss of approximately 0.0001, compared to significantly higher values for SuPerFed (0.0008), FedPer (0.0017), SCAFFOLD (0.0031), and FedProx (0.004). The advantage becomes even more pronounced in risk-sensitive metrics such as VaR95% and CVaR95%, where FLARE-LA achieves much lower values, highlighting its ability to effectively manage tail risks even with limited trading platforms participation. These trends persist in Fig. 2(b), with 20 training rounds further consolidating FLARE-LA's dominance in all metrics.

When the participation rate increases to 30% as shown in Figs. 2(c) and 2(d), the overall model performance improves across all methods. However, FLARE-LA retains a clear advantage, achieving substantially lower Mean Loss values. For instance, in Fig. 2(d), FLARE-LA reaches a Mean Loss of approximately 0.00005, outperforming SuPerFed (0.0003), FedPer (0.0005), SCAFFOLD (0.0010), and FedProx (0.002). The improvement in FLARE-LA's performance with higher participation rates underscores its ability to fully leverage the increased availability of local data while maintaining its computational efficiency and accuracy.

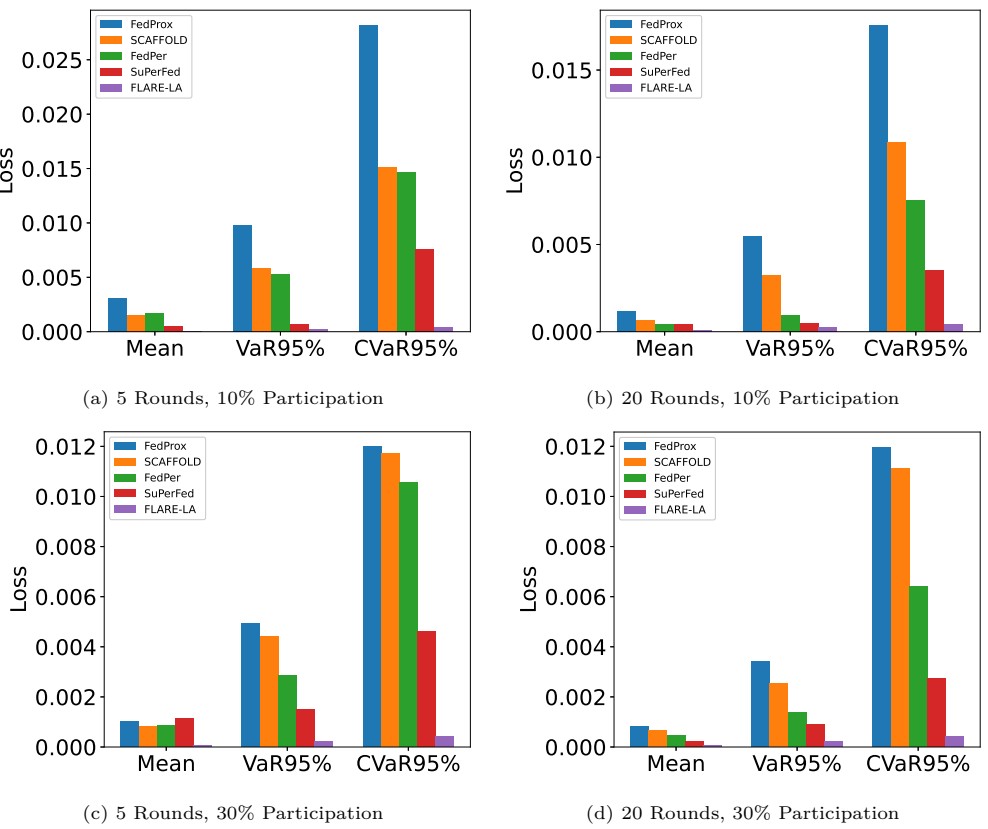

Figure 2: Performance comparison with varying participation rates and training rounds on realized volatility prediction.

Furthermore, the impact of increasing the number of federated training rounds is evident. FLARE-LA demonstrates rapid convergence to low loss values within a few rounds, significantly reducing the computational burden compared to other methods. Even after just 5 training rounds, FLARE-LA achieves results comparable to or better than the baseline methods after 20 rounds, as shown in Figs. 2(b) and 2(d). This efficiency highlights FLARE-LA's capability to deliver robust performance even in scenarios with limited training rounds or participation rates.

As shown in Fig. 3, we provide the comparative analysis across various participation distributions to evaluate the efficacy of FLARE-LA in addressing the inherent challenges of FL with dynamic participation. The experiments were conducted with a 20% participation rate, 10 federated rounds, and 10 local epochs in each round. As shown in Fig. 3(a) where trading platforms are sampled from an exponential distribution, with a scale parameter of 1.0, FLARE-LA demonstrates a remarkable ability to achieve a mean loss of $7.358 \times 10^{-5}$, VaR95% of $2.284 \times 10^{-4}$, and CVaR95% of $3.978 \times 10^{-4}$, outperforming FedProx, SCAFFOLD, and FedPer by an order. As shown in Fig. 3(b), where trading platforms are sampled from a geometric distribution with a probability of success of an individual trial set at 0.35, FLARE-LA once again emerges as the top-performing algorithm. Fig. 3(c) explores the performance of algorithms when trading platforms are sampled from a Gamma distribution, with a shape parameter of 2.0 and a scale parameter of 1.0. In Fig. 3(d), where trading platforms are sampled from a chi-square distribution with the number of degrees of freedom set at 2.0, FLARE-LA continues to outshine the baseline algorithms.

By consistently delivering superior performance metrics, FLARE-LA showcases its adaptability in scenarios characterized by varying levels of data availability and participation. By consistently achieving lower mean loss, VaR95%, and CVaR95% values, FLARE-LA underscores its resilience and adaptability in optimizing federated model training across a spectrum of trading platform distributions. These results demonstrate FLARE-LA's capability in managing volatile market conditions and optimizing federated model training despite unpredictable trading platform participation patterns.

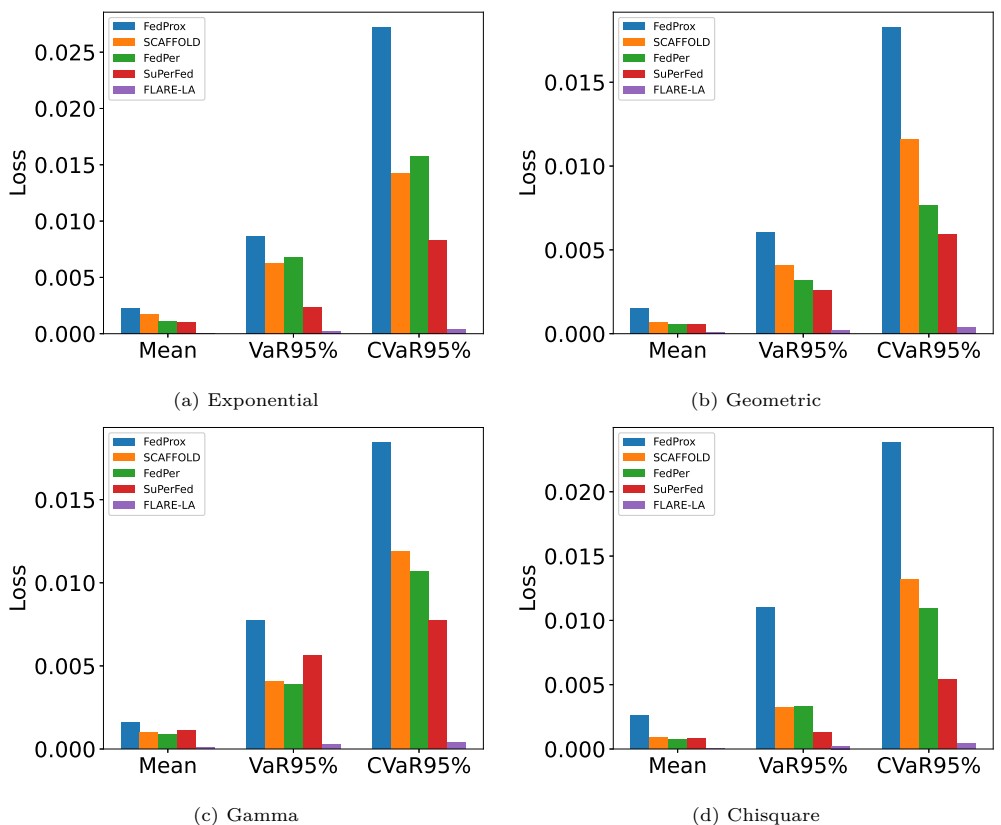

Figure 3: Illustrations of distributions: (a) Exponential, (b) Geometric, (c) Gamma, and (d) Chi-square.

Table 1: The Computation Cost Comparison for One Round

| Participation Rate | Fedprox (s) | SCAFFOLD (s) | FedPer (s) | SuPerFed (s) | FLARE-LA (s) |
|---|---|---|---|---|---|
| 30% | 74.39 | 105.9 | 46.18 | 99.63 | 48.16 |
| 60% | 144.83 | 208.51 | 90.42 | 186.67 | 94.43 |

### 5.1.2 Computation Cost Comparison

The computation cost comparison for one round of FL is presented in Table 1, showcasing FLARE-LA's computational efficiency across varying participation rates. All experiments were conducted on an experimental platform featuring an 8-core CPU, a 14-core GPU, and 16GB of RAM. This setup ensures consistent benchmarking across all evaluated FL methods.

At a participation rate of 30%, FLARE-LA achieves a computation time of 48.16 seconds, significantly outperforming SCAFFOLD (105.9 seconds) and SuPerFed (99.63 seconds). While FedPer demonstrates a slightly faster computation time of 46.18 seconds, its slower convergence rate necessitates more training rounds to achieve comparable results, thereby increasing the overall computational burden. FLARE-LA's superior balance between computational demands and model accuracy ensures efficient and timely model updates, even under challenging participation scenarios.

When the participation rate increases to 60%, FLARE-LA continues to excel with a computation time of 94.43 seconds, outperforming Fedprox (144.83 seconds) and SCAFFOLD (208.51 seconds) by substantial margins. Although FedPer achieves a comparable time of 90.42 seconds, FLARE-LA's faster convergence significantly reduces the total training cost, making it a more efficient and scalable solution for large-scale FL applications in financial markets.

The experimental results demonstrate FLARE-LA's robustness and adaptability in addressing challenges such as fragmented datasets and irregular client participation. By achieving competitive computation times, FLARE-LA ensures privacy-preserving collaboration and timely model updates, addressing critical requirements for FL in decentralized financial environments. Its ability to provide reliable volatility predictions is particularly valuable for effective risk management and investment decision-making in dynamic financial markets.

## 5.2 Experiments with CIFAR10 and MNIST

To extend our evaluation beyond the financial domain, we conducted experiments using the CIFAR10 and MNIST datasets, partitioned into 1000 clients. These datasets serve as benchmarks to demonstrate the generalizability and robustness of FLARE-LA in broader FL scenarios.

To simulate real-world challenges, we introduced artificial label noise into the training sets, employing two commonly used noise schemes: pairwise flipping (Han et al., 2018) and symmetric flipping (Van Rooyen et al., 2015). The pairwise flipping scheme models scenarios where labels transition to semantically similar neighboring labels with a noise ratio $\epsilon$, while retaining the correct label with a probability of $1 - \epsilon$. The symmetric flipping scheme assumes uniform mislabeling across all incorrect labels, distributing the noise ratio $\epsilon$ evenly among them, while preserving the correct label with a probability of $1 - \epsilon$.

For both schemes, the test sets remain clean to ensure a fair and accurate evaluation of model performance. This setup allows us to rigorously assess FLARE-LA's ability to handle noisy labels, which is a critical capability for FL applications in dynamic and unpredictable environments. By addressing label noise effectively, FLARE-LA demonstrates its robustness and adaptability in diverse scenarios, further validating its utility across domains.

### 5.2.1 Convergence Analysis

We assessed the convergence performance of FLARE-LA compared to state-of-the-art FL methods using the CIFAR10 and MNIST datasets. These experiments evaluated FLARE-LA's ability to handle FL challenges, such as non-IID data distributions, dynamic participation, and prolonged training, while maintaining robust convergence across various training rounds. Model performance was measured in terms of Mean Loss, VaR95%, and CVaR95%.

On the CIFAR10 dataset, as shown in Fig. 4, FLARE-LA consistently outperformed the baselines, including FedRep, Ditto, and SuPerFed, across all training stages. At 300 training rounds as illustrated in Fig. 4(a), FLARE-LA achieved a Mean Loss of approximately 1.2, significantly lower than SuPerFed with 2.1, Ditto with 2.7, and FedRep with 4.3, while demonstrating superior tail-risk management with reduced VaR95% and CVaR95%. As training progressed to 400, 500, and 600 rounds as shown in Figs. 4(b)(c)(d), FLARE-LA maintained its dominance, achieving a final Mean Loss of 0.8, solidifying its position as the most robust and reliable method. Compared to the baselines, which struggled with slower convergence and less effective local adaptation, FLARE-LA demonstrated faster convergence and superior overall performance.

We also compare their performance on the MNIST dataset as shown in Fig. 5, FLARE-LA outperformed LG-FedAvg, pFedMe, and FedRep across all training rounds. As shown in Fig. 5(a) with 300 rounds, FLARE-LA achieved a Mean Loss of 0.3, compared to 1.0 for LG-FedAvg and 0.8 for FedRep, while also demonstrating lower VaR95% and CVaR95%, reflecting its superior risk management. As training progressed to 400 and 500 rounds as shown in Figs. 5(b) and 5(c), FLARE-LA's advantage became more pronounced, achieving a Mean Loss of 0.2 by 500 rounds, compared to 0.7 for LG-FedAvg and 0.6 for FedRep. Even as pFedMe slightly narrowed the gap, it failed to match FLARE-LA's efficiency in managing tail risks. By 600 training rounds as shown in Fig. 5(d), FLARE-LA achieved a Mean Loss of 0.1, significantly outperforming FedRep with 0.5 and LG-FedAvg with 0.6, while maintaining lower VaR95% and CVaR95%. These results shown in both Fig. 4 and Fig. 5 highlight FLARE-LA's scalability and adaptability, demonstrating faster convergence and superior performance across diverse FL settings.

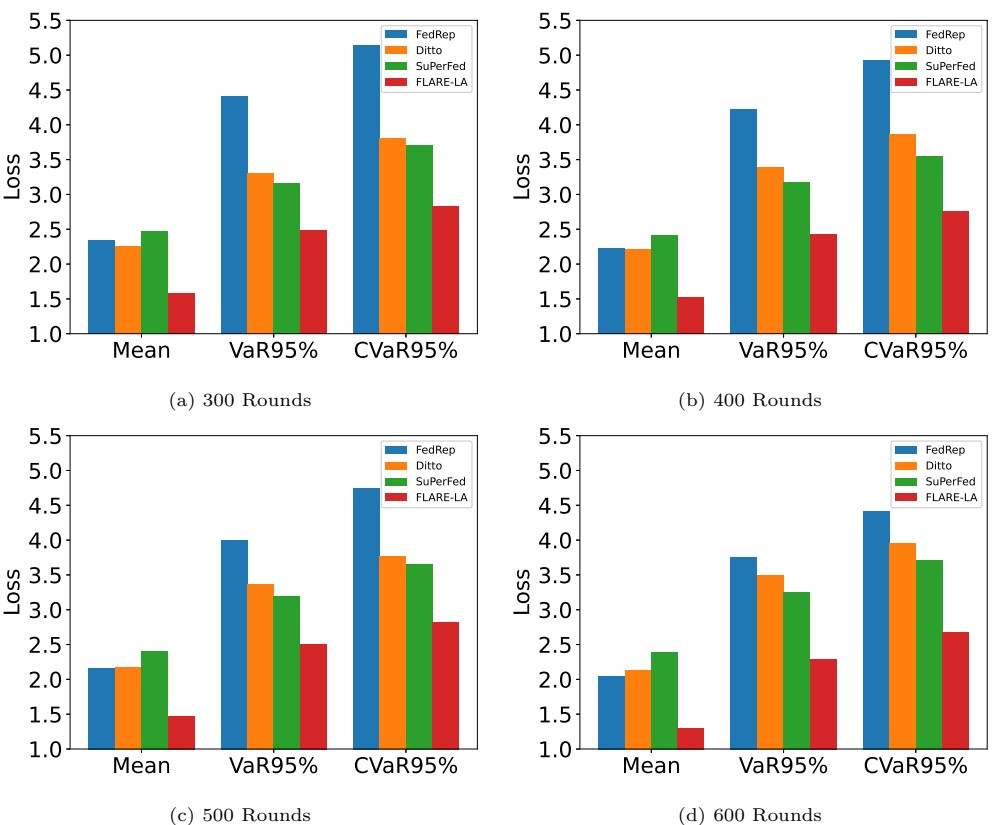

Figure 4: Convergence analysis of model performance across different federated training rounds using the CIFAR10 dataset.

### 5.2.2 Analysis of Label Noise

We evaluated the robustness of FLARE-LA under symmetric and pairwise label noise conditions using the CIFAR10 and MNIST datasets. These experiments tested FLARE-LA's ability to maintain strong performance under varying levels of label noise, comparing it to several baseline methods in terms of Mean Loss, VaR95%, and CVaR95%.

As shown in Fig. 6, FLARE-LA consistently outperformed FedRep, Ditto, and SuPerFed across noise ratios ranging from 0.2 to 0.8 on the CIFAR10 dataset. At a noise ratio of 0.2 as shown in Fig. 6(a), FLARE-LA achieved a Mean Loss of approximately 1.5, significantly lower than SuPerFed with 2.3, Ditto with 2.0, and FedRep with 3.2. As noise levels increased to 0.4 and 0.6 as illustrated in Figs. 6(b) and 6(c), FLARE-LA maintained its competitive edge, with lower losses and risk metrics than the baselines, showcasing its resilience to intermediate noise levels. Even under extreme noise at 0.8 as shown in Fig. 6(d), FLARE-LA demonstrated remarkable robustness, maintaining strong performance, while the baselines exhibited significant degradation, emphasizing FLARE-LA's superior adaptability.

As shown in Fig. 7, FLARE-LA demonstrated similar robustness against symmetric label noise, outperforming LG-FedAvg, pFedMe, and FedRep across all noise levels on the MNIST dataset. At a noise ratio of 0.2, FLARE-LA effectively mitigated the adverse effects of noise, achieving superior predictive accuracy and risk management. As noise levels increased to 0.4, 0.6, and 0.8, FLARE-LA maintained a substantial performance advantage, with significantly lower losses and risk metrics, highlighting its scalability and effectiveness in handling challenging scenarios with noisy labels.

Under pairwise label noise conditions on the CIFAR10 dataset, as shown in Fig. 8, FLARE-LA continued to demonstrate resilience. At a noise ratio of 0.2 illustrated in Fig. 8(a), FLARE-LA achieved a Mean Loss of 2.2, outperforming SuPerFed with 2.5, Ditto with 2.5, and FedRep with 3.0, along with significantly lower VaR95%

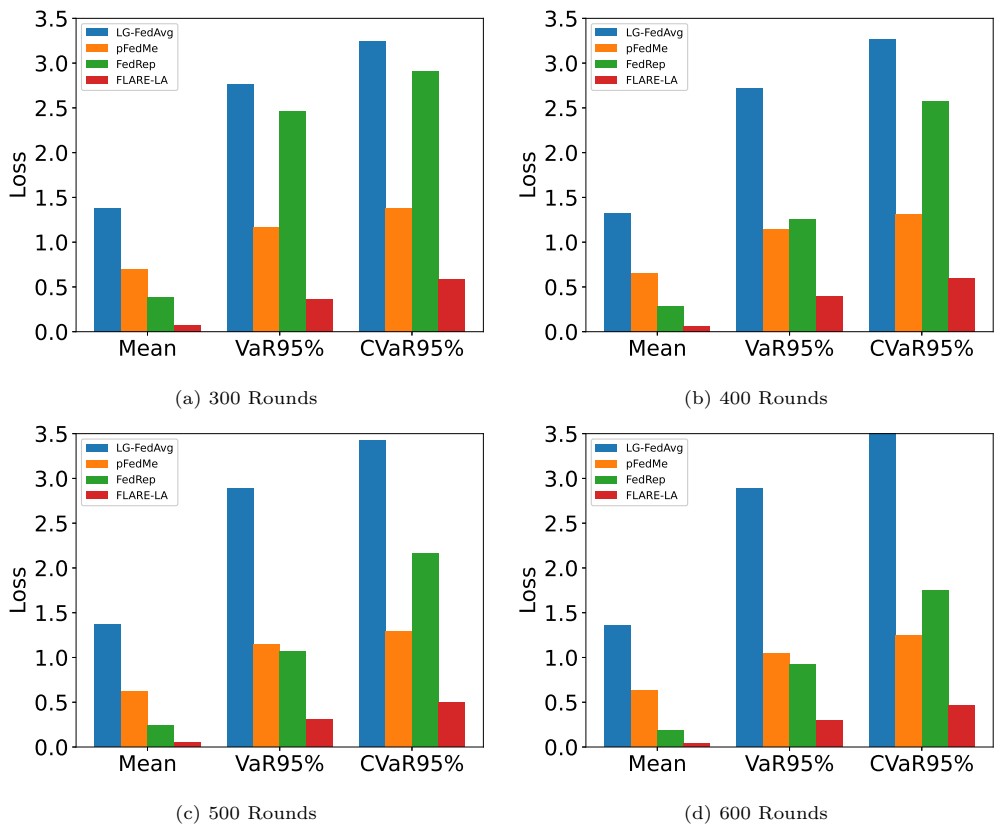

Figure 5: Convergence analysis of model performance across different federated training rounds using the MNIST dataset.

and CVaR95%. At a higher noise ratio of 0.6 as shown in Fig. 8(b), FLARE-LA maintained its superiority, achieving a CVaR95% of 3.5, compared to 3.8 for SuPerFed, 4.1 for Ditto, and 6.6 for FedRep. These results emphasize the efficacy of FLARE-LA's probabilistic local adaptation mechanism in mitigating noise effects and leveraging the structure of local data distributions. FLARE-LA consistently demonstrated superior resilience to label noise across both symmetric and pairwise scenarios, outperforming baseline methods in all metrics. Its robust local adaptation mechanism ensures stable and accurate performance under challenging noisy label conditions, making it a reliable solution in real-world settings.

### 5.2.3 Analysis of Local Data Heterogeneity

We analyzed the impact of local data heterogeneity on the performance of FLARE-LA compared to baseline methods using the CIFAR10 and MNIST datasets. Data heterogeneity was controlled through the Dirichlet distribution's concentration parameter $\alpha$, where smaller $\alpha$ values indicate higher heterogeneity, i.e., non-IID data, and larger $\alpha$ values correspond to more homogeneous distributions, i.e., near-IID data.

The performance compared on the CIFAR10 dataset is shown in Fig. 9, where FLARE-LA consistently outperformed FedRep, Ditto, and SuPerFed across both heterogeneity settings. Under the moderately heterogeneous scenario with $\alpha = 10.0$ as shown in Fig. 9(a), FLARE-LA achieved a Mean Loss of approximately 1.2, significantly lower than SuPerFed, Ditto, and FedRep, all of which reported values above 2.5. FLARE-LA also achieved lower VaR95% and CVaR95%, demonstrating its superior ability to handle diverse data distributions and mitigate tail risks. In the near-IID scenario with $\alpha = 1000.0$ as shown in Fig. 9(b), all methods improved due to reduced heterogeneity, but FLARE-LA maintained its performance advantage, achieving a Mean Loss of 1.5, compared to 2.6 for SuPerFed and Ditto, and 3.0 for FedRep. These results highlight FLARE-LA's adaptability across varying levels of heterogeneity, consistently delivering robust performance.

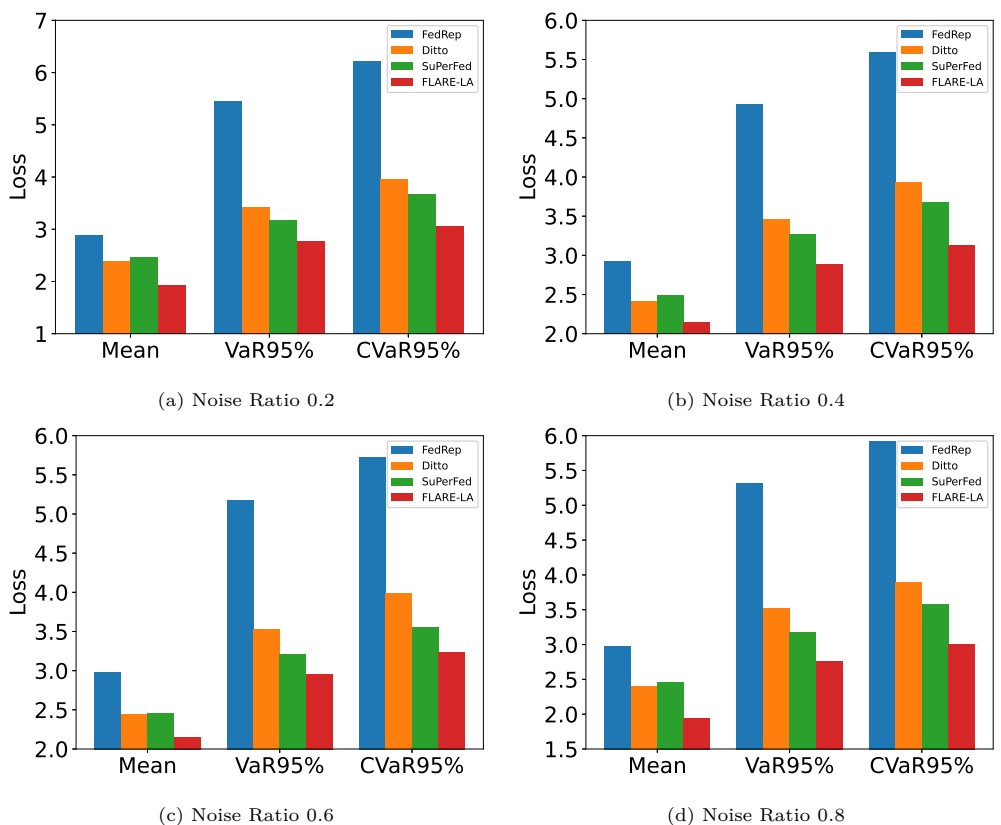

Figure 6: Impact of varying symmetric label noise rates on model performance during federated training with the CIFAR10 dataset.

When compare their performance on the MNIST dataset as shown in Fig. 10, FLARE-LA demonstrated similar dominance over LG-FedAvg, pFedMe, and FedRep. In the highly heterogeneous scenario with $\alpha = 10.0$ illustrated in Fig. 10(a), FLARE-LA achieved the lowest Mean Loss, significantly outperforming the baselines while also maintaining lower VaR95% and CVaR95%, reflecting its superior risk management capabilities. The baselines struggled to generalize effectively under these conditions, with LG-FedAvg and FedRep showing higher losses and slower convergence. Although pFedMe performed competitively, it lagged behind FLARE-LA in accuracy and convergence speed. In the near-IID scenario with $\alpha = 1000.0$ as shown in Fig. 10(b), the performance gap narrowed due to the more uniform data distribution. However, FLARE-LA maintained its advantage, achieving the best results in terms of Mean Loss and risk metrics, while also demonstrating computational efficiency by requiring fewer training rounds to achieve convergence. These results shown in both Fig. 9 and Fig. 10 underscore FLARE-LA's ability to adapt to varying levels of data heterogeneity, consistently outperforming baseline methods across diverse FL scenarios. Its probabilistic local adaptation mechanism and Taylor-based linearization effectively balance global and local contributions, ensuring robust and efficient performance in both non-IID and IID-like settings.

### 5.2.4 Analysis of Client Participation Rates

Then, we analyzed the impact of varying client participation rates on the performance of FLARE-LA compared to baseline methods using the CIFAR10 and MNIST datasets. These experiments simulate dynamic participation scenarios where only a small subset of clients is involved in each training round, evaluating FLARE-LA's robustness under extreme constraints.

On the CIFAR10 dataset, as shown in Fig. 11, two participation rates were tested, i.e., 1% and 2%. At the 1% participation rate as illustrated in Fig. 11(a), FLARE-LA demonstrated exceptional robustness and stability, significantly outperforming FedRep, Ditto, and SuPerFed in both convergence speed and predictive

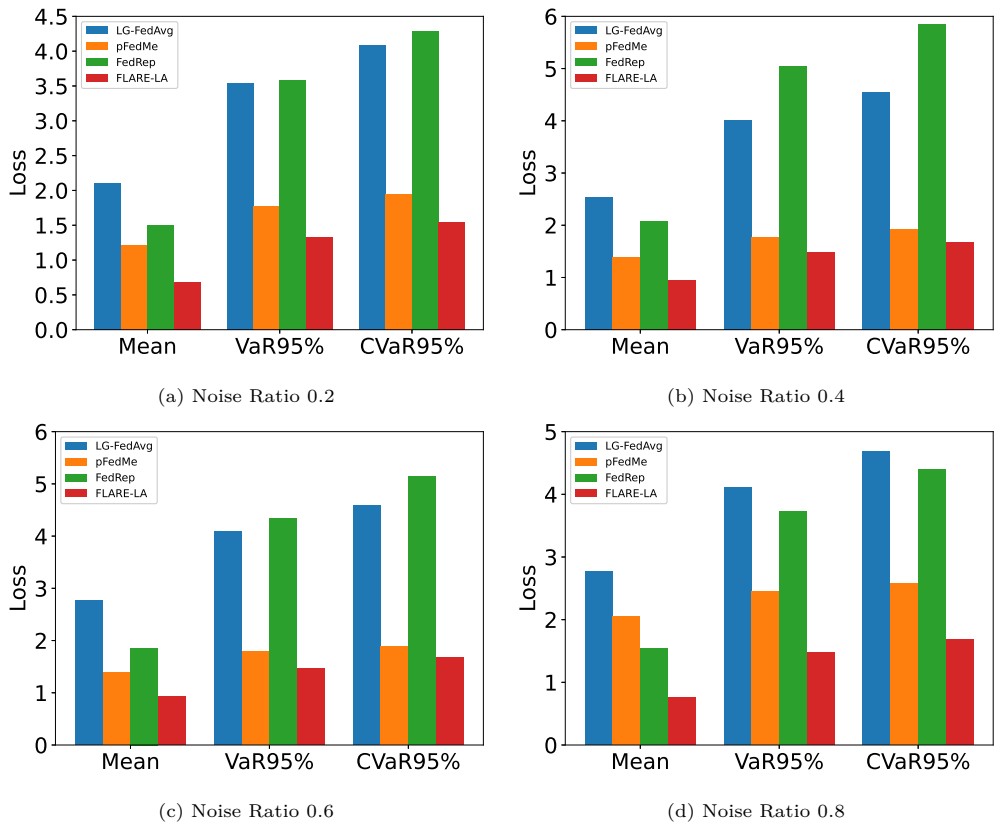

(a) Noise Ratio 0.2

(b) Noise Ratio 0.4

(c) Noise Ratio 0.6

(d) Noise Ratio 0.8

Figure 7: Impact of varying symmetric label noise rates on model performance during federated training with the MNIST dataset.

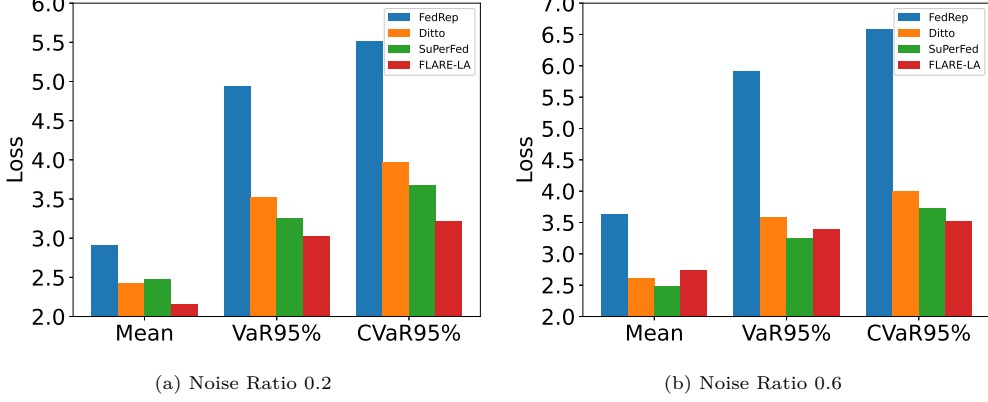

(a) Noise Ratio 0.2

(b) Noise Ratio 0.6

Figure 8: Impact of varying pairwise label noise rates on model performance during federated training with the CIFAR10 dataset.

accuracy. The probabilistic local adaptation mechanism in FLARE-LA ensures effective personalization while preserving generalization, even with limited client engagement. At the 2% participation rate as shown in Fig. 11(b), all methods improved due to increased client involvement, but FLARE-LA maintained its advantage with faster convergence and superior performance metrics.

As shown in Fig. 12, four participation rates were tested, i.e., 0.5%, 1%, 1.5%, and 2.0% on the MNIST dataset. At the lowest participation rate of 0.5% as illustrated in Fig. 12(a), FLARE-LA achieved a mean loss of approximately 0.2, significantly outperforming LG-FedAvg with 1.4, pFedMe with 0.6, and FedRep with 0.4. Its risk metrics, VaR95% and CVaR95%, were consistently lower than those of the baselines, reflecting superior

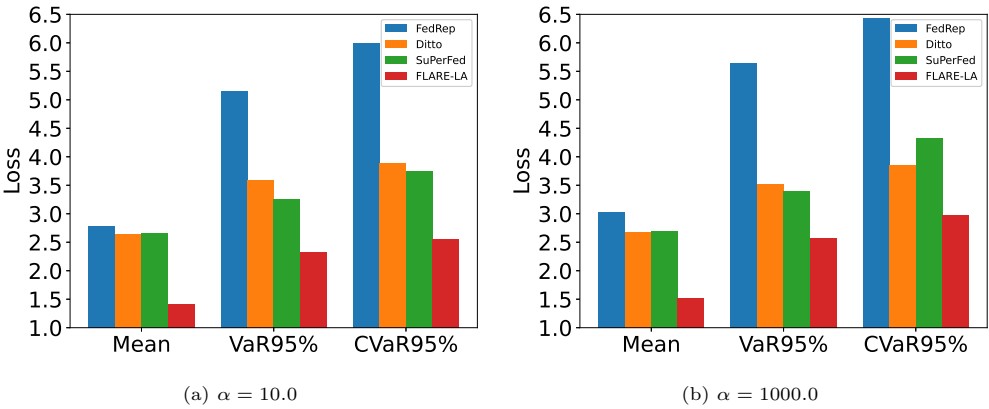

Figure 9: Impact of varying degrees of local data heterogeneity on model performance during federated training with the CIFAR10 dataset.

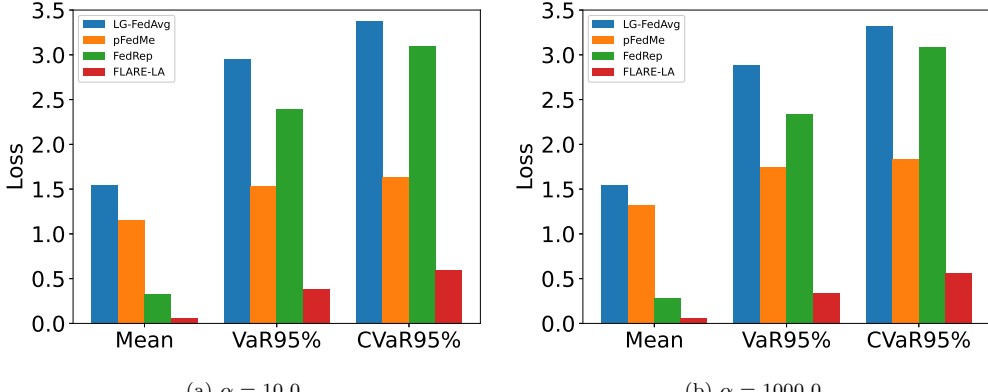

Figure 10: Impact of varying degrees of local data heterogeneity on model performance during federated training with the MNIST dataset.

robustness. As participation rates increased to 1%, 1.5%, and 2.0% shown in Figs. 12(b)(c)(d), FLARE-LA maintained its performance lead, achieving the lowest mean loss and risk metrics across all settings. At the highest participation rate of 2.0%, FLARE-LA effectively leveraged increased client engagement to achieve the most favorable results, further underscoring its scalability and adaptability. The above results highlight FLARE-LA's ability to maintain robustness and generalization across dynamic participation scenarios, consistently outperforming baseline methods. Its efficient local adaptation mechanism ensures stable and accurate performance, even under extreme constraints or varying client participation rates.

## 5.3 Performance Comparison with Additional Local Updates

To further highlight the superiority of FLARE-LA in adaptive local updating, beyond the innovative two-stage framework, we conducted a comparative analysis by equipping existing baseline methods with additional local updates following the FL training phase. Specifically, we modified LG-FedAvg (Liang et al., 2020), APFL (Deng et al., 2020), FedRep (Collins et al., 2021), FedPer (Arivazhagan et al., 2019), Ditto (Li et al., 2021), and SuPerFed (Hahn et al., 2022) to include this adjustment, resulting in the variants LG-FedAvg-LE, APFL-LE, FedRep-LE, FedPer-LE, Ditto-LE, and SuPerFed-LE.

The purpose of this comparison is to demonstrate that FLARE-LA's design offers significant advantages over these locally enhanced baselines, even when they incorporate additional efforts to improve local performance. Unlike these approaches, FLARE-LA is explicitly designed to address the trade-off between global knowledge preservation and localized adaptation, ensuring both robustness and generalization across heterogeneous

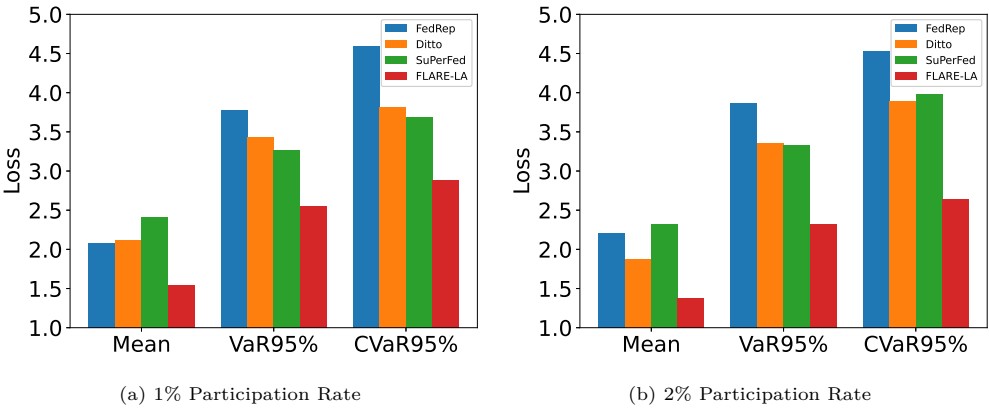

(a) 1% Participation Rate

(b) 2% Participation Rate

Figure 11: Impact of varying client participation rates on model performance during federated training with the CIFAR10 dataset.

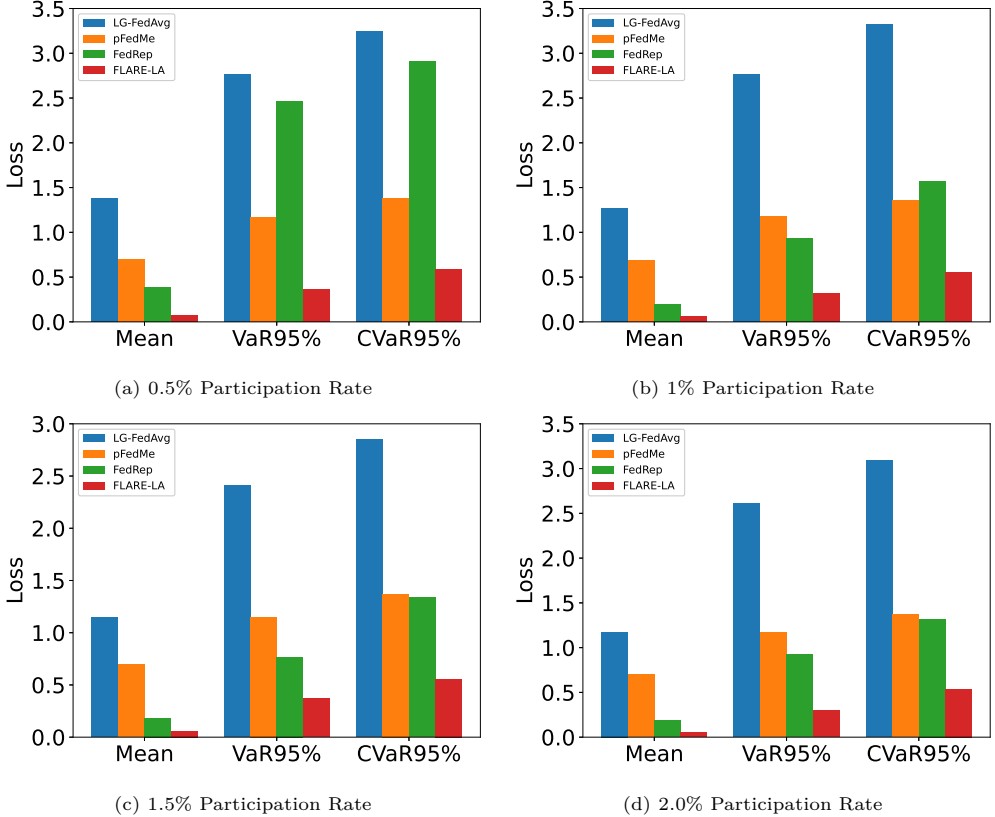

(a) 0.5% Participation Rate

(b) 1% Participation Rate

(c) 1.5% Participation Rate

(d) 2.0% Participation Rate

Figure 12: Impact of varying client participation rates on model performance during federated training with the MNIST dataset.

datasets. The experiments, conducted on multiple datasets, show that FLARE-LA achieves superior performance without sacrificing the benefits of the collaborative FL training.

### 5.3.1 Performance Comparison on the Volatility Prediction Dataset

We evaluated the performance of FLARE-LA against locally enhanced baselines LG-FedAvg-LE, Ditto-LE, APFL-LE, and FedPer-LE using the volatility prediction dataset. The experiments involved 20% of 10,000

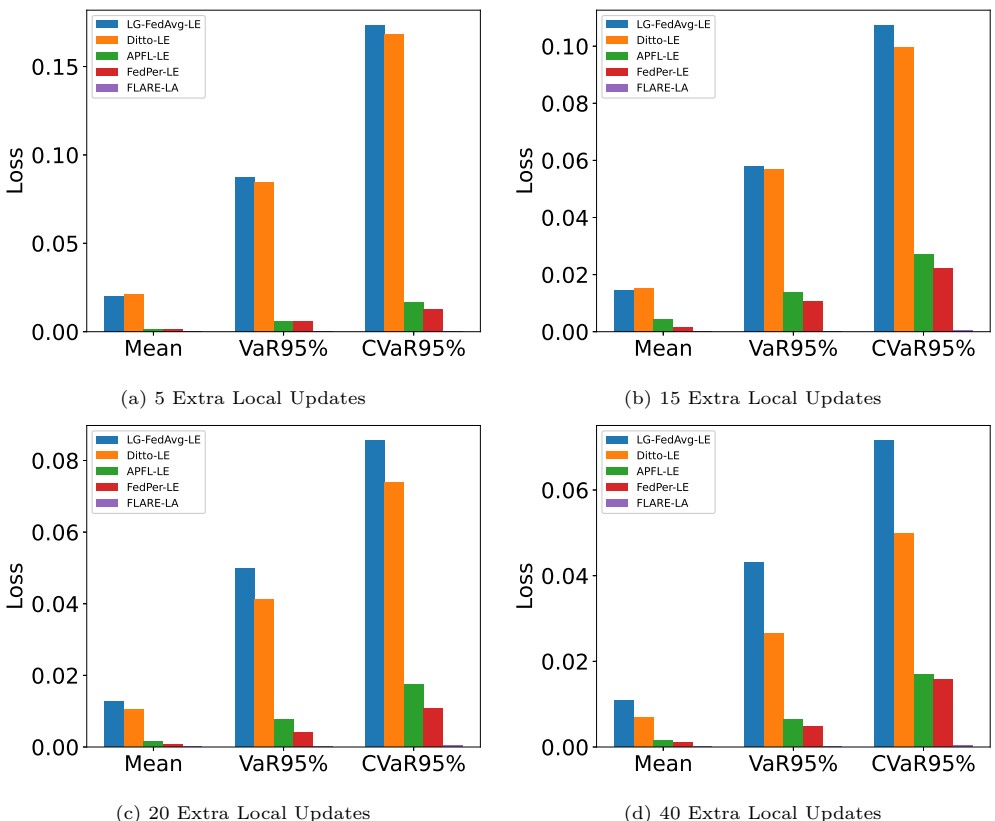

(a) 5 Extra Local Updates

(b) 15 Extra Local Updates

(c) 20 Extra Local Updates

(d) 40 Extra Local Updates

Figure 13: Performance comparison of FLARE-LA against FL baselines enhanced with increasing numbers of local updates after the FL training procedure on the volatility prediction data set.

trading platforms, with 10 rounds of FL training and 10 local epochs per round. We varied the number of additional local updates applied to the baselines, increasing from 5 in Fig. 13(a) to 40 in Fig. 13(d).

The results demonstrate a clear and consistent advantage of FLARE-LA over the locally enhanced baselines, which achieves a significantly lower mean loss and risk metrics compared to all baselines. As shown in Fig. 13(a), FLARE-LA's mean loss is an order of magnitude lower than the best-performing baseline, FedPer-LE, and its risk values, i.e., VaR95% and CVaR95%, are reduced by more than 95%. These results highlight FLARE-LA's ability to effectively balance global knowledge and localized adaptation, avoiding the overfitting and drift that additional local updates can introduce.

As the number of local updates increases from 5 to 40, the baselines show an expected improvement in personalization, reflected in slightly lower mean losses. However, this comes at a significant cost to generalization and robustness. For instance, while baselines like APFL-LE and FedPer-LE demonstrate marginal improvements in their mean loss with more local updates, their VaR95% and CVaR95% values degrade, indicating a higher risk of poor performance on extreme cases. These results confirm the effectiveness of FLARE-LA's two-stage framework. While the locally enhanced baselines attempt to improve personalization through additional updates, they cannot balance between personalization, robustness, and generalization. The proposed Taylor-based linearization and probabilistic local adaptation mechanisms in FLARE-LA ensure that local updates enhance model performance without deviating from the collaborative objectives of federated training efforts. Therefore, FLARE-LA achieves superior performance in both mean loss and risk metrics, making it a more reliable and efficient solution for volatility prediction tasks.

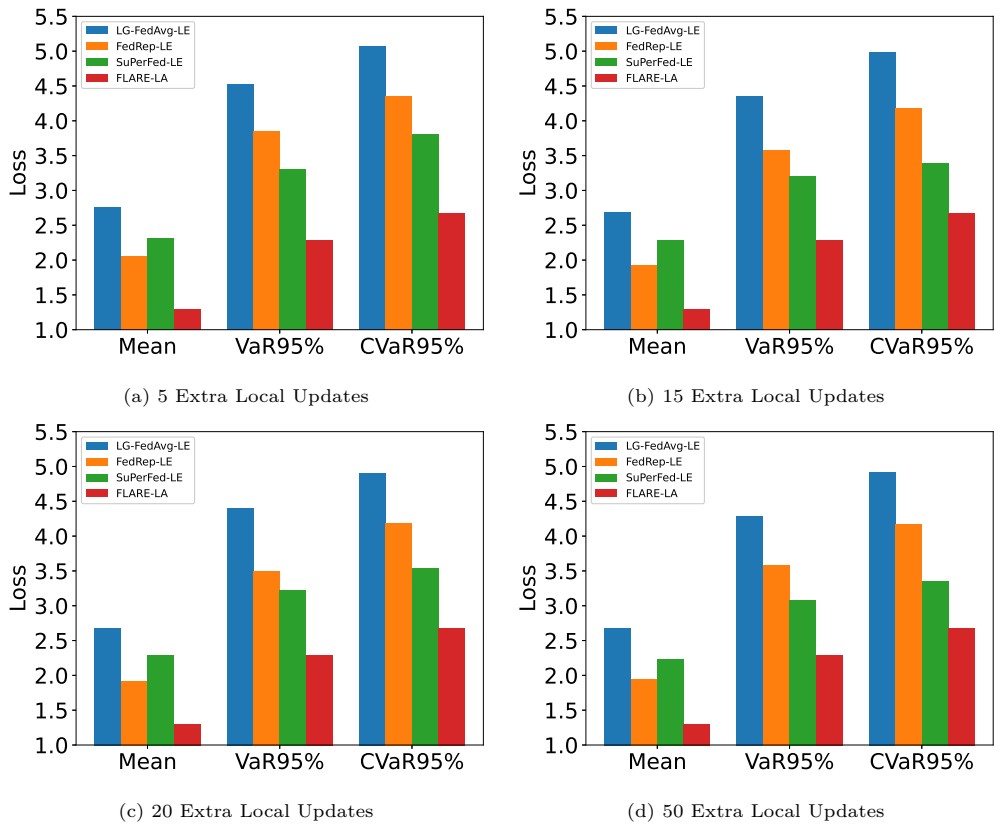

Figure 14: Performance comparison of FLARE-LA against FL baselines enhanced with increasing numbers of local updates after the FL training procedure on the CIFAR10 dataset.

### 5.3.2 Performance Comparison on the CIFAR10 and MNIST Datasets

Furthermore, we also evaluated the performance of FLARE-LA against the extra local updating enhanced baselines using the CIFAR10 and MNIST datasets. Both experiments involved 600 federated training rounds, each with 5 local epochs, followed by additional local updates for the baselines, ranging from 5 to 50. Across all settings, FLARE-LA consistently outperformed the baselines in terms of both mean test loss and risk metrics, i.e., VaR95% and CVaR95%, demonstrating its robustness and generalization capabilities.

As shown in Fig. 14, FLARE-LA achieved a mean test loss of 1.2928 and CVaR95% of 2.6734, remaining stable across all local update settings on the CIFAR10 dataset. In contrast, the baselines showed slight improvements in mean test loss with additional local updates but exhibited significant inconsistencies in risk metrics. Even with 50 updates, FedRep-LE achieved a mean loss of 1.9511 but had a CVaR95% of 4.1711, highlighting its lack of robustness compared to FLARE-LA.

On the MNIST dataset, as shown in Fig. 15, FLARE-LA demonstrated even greater stability, achieving a mean test loss of 0.0427 and a CVaR95% of 0.4679, irrespective of the number of local updates applied to the baselines. By contrast, the baselines showed considerable variability, with FedRep-LE achieving its best mean test loss of 0.1745 at 50 updates, but its CVaR95% remained high at 1.2898. Similar trends were observed with LG-FedAvg-LE and Ditto-LE, which failed to consistently balance local personalization and global generalization, leading to unstable risk metrics.

The results as shown in both Fig. 14 and Fig. 15 demonstrate that FLARE-LA effectively integrates global and local knowledge, maintaining superior performance without requiring additional local updates. The stability and robustness of FLARE-LA, as evidenced by consistently low risk metrics across both datasets, highlight the effectiveness of its Taylor-based linearization and probabilistic local adaptation mechanisms. In

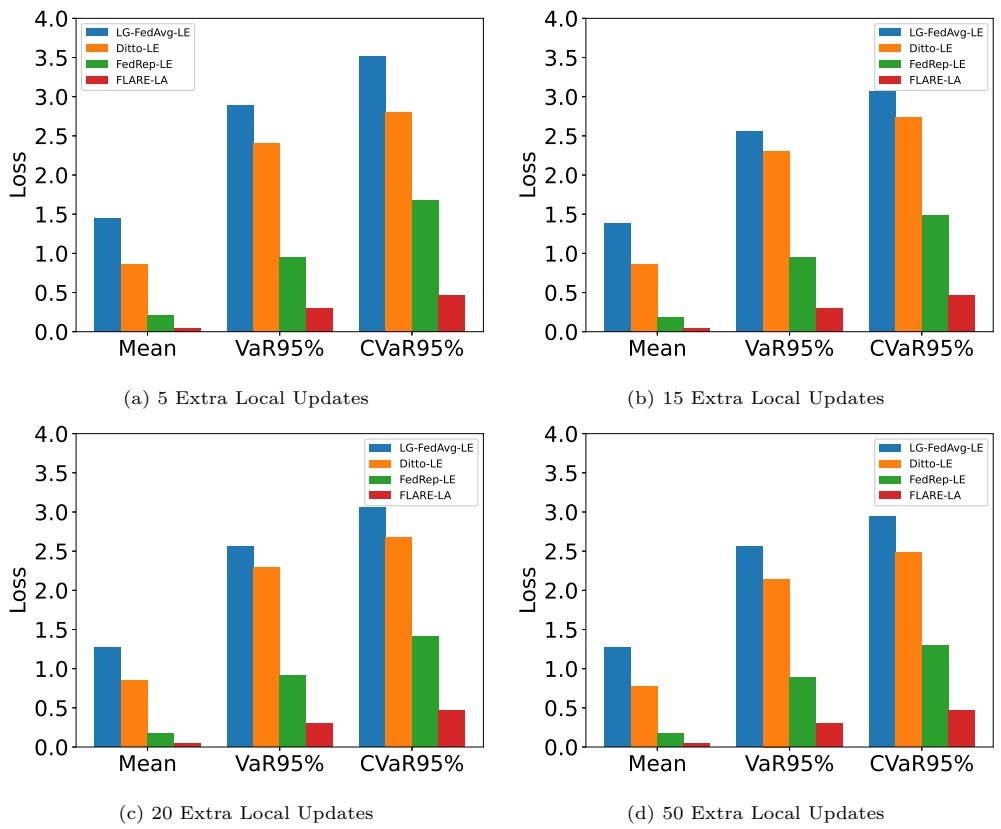

(a) 5 Extra Local Updates

(b) 15 Extra Local Updates

(c) 20 Extra Local Updates

(d) 50 Extra Local Updates

Figure 15: Performance comparison of FLARE-LA against FL baselines enhanced with increasing numbers of local updates after the FL training procedure on the MNIST dataset.

contrast, the baselines struggled with significant variability and reduced generalization, even when enhanced with increasing local updates.

# 6 Conclusions and Discussions

This work proposed FLARE-LA, a novel framework addressing the challenges of FL in diverse and dynamic environments, particularly financial markets. By integrating Taylor-based linearization for efficient local adaptation with a probabilistic mechanism leveraging the Jacobian matrix, FLARE-LA achieves precise optimization, robust performance, and interpretable uncertainty quantification. Extensive experiments demonstrated its superiority over state-of-the-art baselines, achieving higher accuracy, faster convergence, and resilience to label noise and dynamic participation. In financial applications, FLARE-LA excelled in metrics like mean loss, VaR95%, and CVaR95%, underscoring its suitability for high-stakes, heterogeneous environments. With its ability to adapt global models to local distributions, handle fragmented datasets, and ensure computational efficiency, FLARE-LA offers a scalable and versatile solution for FL challenges. Future directions include extending the framework to domains like healthcare and IoT, integrating advanced optimization techniques, positioning FLARE-LA as a foundation for advancing FL innovations.

**Acknowledgments**

This work was supported in part by the Natural Sciences and Engineering Research Council of Canada and Compute Canada.

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
