# OpenReview forum: "Federated Learning with Efficient Local Adaptation for Realized Volatility Prediction"
_TMLR — Accepted by TMLR_

### Review · Reviewer_ciru · 2024-10-26

**Summary Of Contributions:**

The paper focus on a specific situation of financial market, which may have isolated datasets. Federated learning (FL) can optimize over all dataset, but may lack performance in some of the local dataset. This paper present Federated Learning with Efficient Local Adaptation (FLELA), which can make a local adaptation step after global optimization. The method first do global optimization based on square loss, then linearize the model at every local dataset and chose the optimal local minimum for every local dataset, using maximum likelihood loss. The algorithm performs good on various dataset and method.

**Audience:**

Yes

**Broader Impact Concerns:**

This paper’s focus on improving local adaptation in federated learning has potential benefits for privacy and accuracy in financial markets, where data heterogeneity is prevalent. It can significantly improve the local performance of FL, and help speed up the time comparing to alternative methods. However, relying on linear approximations may limit its effectiveness in capturing complex local patterns, particularly in diverse datasets.

**Claims And Evidence:**

Yes

**Requested Changes:**

I would like to request experiment on comparison for fine-tuning the model on each dataset. And also some case that linear function cannot capture the local model efficiently. Also for the paper writing I would suggest to make the figure smaller and dense(not to put it in one page).

**Strengths And Weaknesses:**

The paper focus on improving local performance for Federated learning. To be honest, I am not an expertise in this area, but I will try my best to help you. It use linearization on the optimal solution in FL part and adapt it into local model, so can easily speed up on time. There are several benefit to do it: Linearization around the optimal solution in FL part is quite easy; we only need to do it once for each local model; it can help improve performance much if the local distribution is different from global. But the way seems a little bit naive to me -- can we just fine tune the model on different dataset to get a better performance for each dataset, or explore the distribution of different local dataset during the global gradient decent step? I am also concerning about whether linear-model is sufficient for a large model -- now the experiment only takes about 40 neurons and that seems to be a small network.

---

> ### Author Response · Authors · 2024-12-11
> **Clarifications on Linearization, Model Size, and Figure Presentation**
>
> Thank you for your detailed feedback and thoughtful questions. Below, we provide a clearer explanation of the novelty and strengths of our method.
>
> The novelty of our method lies in its ability to efficiently improve local adaptation within FL through a combination of Taylor-based linearization, a probabilistic framework, and a scalable computational design. These innovations address key challenges in FL, particularly for applications like financial volatility prediction, where data heterogeneity and dynamic participation are prevalent.
>
> First, our method employs Taylor expansion-based linearization to tailor the global model to each local dataset. By leveraging the Jacobian matrix, we efficiently capture local demands while maintaining the global model's knowledge. Importantly, this linearization is computationally efficient, as it only needs to be performed once per local model, significantly reducing overhead compared to iterative fine-tuning collaborative approaches. This design ensures that even when local distributions diverge from the global model, the adaptation remains precise and interpretable.
>
> Moreover, we frame local adaptation within a probabilistic framework, which enhances robustness by incorporating uncertainty quantification. This is particularly critical for high-stakes applications like financial markets, where understanding prediction confidence is essential. Unlike simple linear models, our approach introduces posterior inference in the function space, making local updates more adaptable and reliable. It is essential to clarify that the linearization in our method is not merely applying a static linear model. Instead, it dynamically adjusts the global model to local distributions by leveraging the structure of the global optimization. This ensures that the adaptation is both efficient and effective, even for larger networks or datasets.
>
> The current popular approaches in personalized FL, as demonstrated by our baseline methods, primarily focus on exploring local dataset distributions during the global gradient descent step. Our proposed method complements these strategies by emphasizing post-FL local adaptation. This approach allows our framework to leverage the strengths of existing FL solutions while further enhancing adaptability. By incorporating this additional step, the linearized local model becomes an enhanced version of the well-optimized global model, offering improved performance and better alignment with client-specific data distributions.
>
> We have also revised the manuscript to make the figures more compact and visually streamlined. We appreciate your suggestions on improving the paper's clarity and structure.

---

### Review · Reviewer_4V13 · 2024-11-01

**Summary Of Contributions:**

The paper proposes Federated Learning with Efficient Local Adaptation (FLELA) to improve realized volatility prediction in decentralized financial markets. Given the data heterogeneity across trading platforms and dynamic participation in FL, the authors developed a novel FL approach combining global model training with locally adaptive updates. FLELA leverages probabilistic frameworks and linearized local adaptations to optimize model predictions for each trading platform. The experimental results show that FLELA outperforms some existing FL methods in both performance and computation time.

**Audience:**

Yes

**Claims And Evidence:**

Yes

**Requested Changes:**

1. Explain why FLELA only focuses on the financial market field.
2. Please test FLELA and other baselines on other datasets.
3. Add more recent personalized and local adaptation FL methods as baselines.

**Strengths And Weaknesses:**

**Strengths:**
1. The authors realize the property of financial markets that is heterogeneous and dynamic in FL.
2. The proposed algorithm, FLELA, uses a local adaptation method to fit such a dynamic system well.
3. The empirical results demonstrate that FLELA outperforms existing FL algorithms in performance (mean loss, VaR, and CVaR) and computation time.

**Weaknesses:**
1. The paper only considers the financial field. The properties of heterogeneity and dynamic participation are common in FL, not just in the financial case. Why not consider a general case and test this algorithm on more datasets/benchmarks?
2. The experiment is on only one dataset and is a bit limited.
3. The chosen baselines don't include recent methods. FedProx is proposed in 2020, and SCAFFOLD and FedPer in 2019.

---

> ### Author Response · Authors · 2024-12-11
> **Broader Applicability Beyond the Financial Domain**
>
> Thank you for your thoughtful comment. While our work uses volatility prediction as the primary application, our algorithm is not limited to the financial domain. Financial markets were chosen as a test case due to their extreme levels of data fragmentation, non-IID data distributions, and stringent privacy concerns, which are more pronounced than in many other domains. These challenges highlight the strengths of our method, particularly its ability to adapt efficiently to rapid changes in local data distributions.
>
> The decision to focus on financial markets also stems from the high stakes and immediate applications of accurate volatility prediction. Financial forecasting tasks demand precise and interpretable models due to the direct monetary implications of prediction errors. FLARE-LA's ability to provide uncertainty estimates and tailored local adaptation addresses these requirements effectively, making the financial domain not only a practical choice but also a domain that fully leverages the framework's strengths.
>
> To address your suggestion, we have extended our experiments to include additional datasets and benchmarks from other domains to demonstrate the broader applicability of our proposed method. Furthermore, we have included additional discussions in the revised manuscript to clarify that while FLARE-LA was originally designed with financial markets in mind, its core principles, i.e., efficient local adaptation and handling dynamic participation, are generalizable and suitable for a wide range of FL contexts.

---

> ### Author Response · Authors · 2024-12-11
> **Expanded Experiments on Additional Datasets**
>
> Thank you for the comment. To address this, we have included experiments on additional publicly available non-financial datasets commonly used in the FL literature, such as the CIFAR-10 and MNIST datasets, which will allow us to demonstrate the effectiveness and generalizability of the proposed method in handling heterogeneity and dynamic participation in diverse contexts.

---

> ### Author Response · Authors · 2024-12-11
> **Inclusion of Recent Baselines for Experimental Comparison**
>
> Thank you for highlighting this important point. In response to your feedback, we have expanded the experimental comparisons in the revised manuscript to include several recent methods specifically designed for personalized and local adaptation in federated learning. These methods include FedProx, SCAFFOLD, FedPer, LG-FedAvg, pFedMe, Ditto, FedRep, and SuPerFed.
>
> The additional comparisons comprehensively evaluate FLARE-LA's performance against these state-of-the-art approaches. The results demonstrate that FLARE-LA consistently outperforms these methods in scenarios requiring robust local adaptation, particularly in environments with high data heterogeneity and dynamic client participation. This expanded evaluation further underscores the effectiveness and adaptability of FLARE-LA.

---

> ### Comment · Reviewer_4V13 · 2024-12-18
>
> Thanks for the author's detailed reply. It addressed my concerns.

---

### Review · Reviewer_fzmC · 2024-11-06

**Summary Of Contributions:**

The paper studies the problem of predicting realized volatility in financial markets using federated learning.  This paper introduces an approach called Federated Learning with Efficient Local Adaptation (FLELA).  By incorporating localized linearization techniques and probabilistic frameworks, FLELA is designed to enhance the fit of a global model to heterogeneous data across different trading platforms. The authors demonstrate that FLELA achieves accurate and efficient predictions.

**Audience:**

Yes

**Broader Impact Concerns:**

No broader impact concerns.

**Claims And Evidence:**

Yes

**Requested Changes:**

1. The authors should discuss why direct fine-tuning on local datasets was not considered as a baseline approach. Including an experimental comparison with this strategy would help to validate the necessity and effectiveness of the proposed method.
2. It is crucial to discuss relevant literature on adaptation and personalization within federated learning, such as references [1, 2, 3, 4].
3. The paper should compare FLELA against other federated learning methods that focus on local adaptation, such as those cited in references [3, 4].

**Strengths And Weaknesses:**

**[Strengths]**
1. The paper studies the critical challenge of dataset heterogeneity in the prediction of financial volatility, a significant barrier in financial analytics.
2. The technique of linearizing neural networks for local adaptation within a federated learning framework represents a reasonable innovation. This approach allows for more precise and efficient prediction of local datasets, improving overall prediction performance.
3. The experimental results demonstrate that FLELA achieves efficient prediction compared to other baselines.

**[Weakness]**
1. The paper does not explore or compare against simpler baseline such as direct fine-tuning of parameter w on local datasets using standard techniques like SGD (by minimizing the prediction error). This omission raises questions about whether the proposed complex method is necessary or optimal.
2. The paper did not discuss several related literature that focus on adaptation strategies in federated learning. Recent literature [1,2,3,4] offers insights into personalized model training within federated frameworks, which are directly relevant to this work.
3. It lacks a comparative analysis with methods specifically designed for enhanced local adaptation within federated learning contexts, such as [3,4]. This comparison is essential to fully evaluate the effectiveness of FLELA in scenarios that demand strong local performance.

[1]  Meng, Lei, et al. "Improving Global Generalization and Local Personalization for Federated Learning." IEEE Transactions on Neural Networks and Learning Systems (2024).
[2] Tan, Alysa Ziying, et al. "Towards personalized federated learning." IEEE transactions on neural networks and learning systems 34.12 (2022): 9587-9603.
[3] Chen, Daoyuan, et al. "Efficient personalized federated learning via sparse model-adaptation." International Conference on Machine Learning. PMLR, 2023.
[4] Yu, Tao, Eugene Bagdasaryan, and Vitaly Shmatikov. "Salvaging federated learning by local adaptation." arXiv preprint arXiv:2002.04758 (2020).

---

> ### Author Response · Authors · 2024-12-11
> **Comparison with Direct Fine-Tuning as a Baseline Approach**
>
> Thank you for your valuable feedback and insightful suggestion. In response, we have incorporated an experimental comparison with the direct fine-tuning approach, which involves optimizing the local parameters on individual datasets. We have included detailed discussions and experimental results in Section 5.1.1 of the revised manuscript to reflect this analysis.
>
> Our results demonstrate that direct fine-tuning falls short due to data heterogeneity or limited local data availability, which are key challenges in FL. Specifically, direct fine-tuning tends to overfit to local datasets and fails to effectively utilize global insights, leading to suboptimal generalization. Our proposed method consistently outperforms direct fine-tuning by efficiently leveraging global knowledge while adapting to local requirements. This balance is achieved through the Taylor-based linearization approach and probabilistic adaptation based on the FL converged global model, which ensure robust performance across diverse datasets.

---

> > ### Comment · Action_Editor_NTWv · 2024-12-24
> >
> > I just want to follow up on this issue, as it's also mentioned by Reviewer ciru in a private comment.
> >
> > It seems that the direct fine-tuning baseline suggested by the reviewers here refers to a two-stage algorithm with a similar structure as the main algorithm of the paper. The only difference is to replace the local adaptation stage (least square) by SGD, which doesn't rely on explicit linearization on the neural network. This is not the same as the Individual Train baseline tested in Section 5.1.1.
> >
> > To strengthen the claims in this paper, I encourage the authors to test this baseline as well.

---

> > > ### Author Response · Authors · 2024-12-31
> > > **The direct fine-tuning baseline**
> > >
> > > Thank you very much for your follow-up and for highlighting the concerns regarding the direct fine-tuning baseline.
> > >
> > > To address this concern, we have implemented and tested the suggested baseline, and the results are included in Section 5.3 of the revised manuscript. This additional comparison provides a more comprehensive evaluation and significantly strengthens the rigor of our analysis, highlighting the clear advantages of FLARE-LA over the baselines. We sincerely appreciate these valuable suggestions, which have helped improve the quality and completeness of our work.
> > >
> > > The proposed FLARE-LA consistently outperforms the locally enhanced baselines by effectively leveraging global knowledge while adapting to local requirements. Unlike the baselines, which rely on additional local updates to improve personalization, FLARE-LA incorporates a Taylor-based linearization strategy and a probabilistic adaptation mechanism. These innovations ensure that local updates remain aligned with the global model, avoiding the common drift observed in federated learning settings with heterogeneous data distributions. Such drift often results in a loss of generalization and robustness, particularly when additional local updates dominate the training process. In contrast, FLARE-LA preserves the collaborative training efforts of the FL stage while enhancing local performance in a balanced and systematic manner.
> > >
> > > The experimental results confirm that while baselines with extra local updates show minor improvements in average test loss, their risk metrics, e.g., VaR95\% and CVaR95\%, demonstrate significant instability, especially as the number of local updates increases. FLARE-LA consistently maintains superior performance across all metrics, including substantially lower VaR95\% and CVaR95\% values, underscoring its robustness in both generalization and worse-case performance. In comparison, the baselines demonstrate performance variability and degradation in risk metrics.
> > >
> > > Simply using the FL-trained global model as an initial point for additional local updates compromises the model's ability to handle information beyond the local datasets. Our method addresses this challenge by maintaining a systematic balance between global and local contributions. This is achieved through the integration of Taylor-based linearization, which enhances the generalization and probabilistic adaptation, ensuring that local updates adjust to specific local needs without sacrificing the applicability of the global model. These carefully designed mechanisms allow FLARE-LA to achieve stability and robustness.

---

> ### Author Response · Authors · 2024-12-11
> **Discussion of Related Literature on FL Adaptation Strategies**
>
> We appreciate the references provided. We have revised the Related Work section to include a discussion of the following.
>
> Our method distinguishes itself by employing Taylor-based linearization and probabilistic adaptation to enable efficient and interpretable local updates, particularly in highly fragmented financial datasets, which complements the insights provided in [1] by offering a more computationally efficient framework for local adaptation. [2] focused on personalized FL by incorporating meta-learning to fine-tune local models and introducing personalized layers for better client-specific adaptations. While their approach effectively handles non-IID data distributions, our method achieves similar goals without requiring additional network components. By leveraging Jacobian-based linearization, we provide a lightweight and scalable mechanism for adapting global models to local data distributions, making it particularly suitable for resource-constrained and swift responding settings in FL applications. [3] introduced sparse model adaptation as an efficient strategy for personalized FL. Their work highlights the benefits of sparsity in improving scalability and computational efficiency during local updates. Similarly, [4] emphasized the importance of robust local adaptation by proposing fallback strategies to mitigate the limitations of global aggregation. Both papers underscore the critical need for effective local updates in federated settings. In contrast, our method advances these concepts by integrating probabilistic modeling with linearized adaptation, ensuring both robustness and scalability. Additionally, our approach directly addresses the challenge of dynamic client participation, which has not been thoroughly addressed in these works.

---

> ### Author Response · Authors · 2024-12-11
> **Comparative Analysis with Enhanced Local Adaptation Methods**
>
> Thank you for your insightful feedback. In response, we have expanded the experimental section to include a comparative analysis with additional baselines specifically designed for enhanced local adaptation in FL contexts, including FedProx, SCAFFOLD, FedPer, LG-FedAvg, pFedMe, Ditto, FedRep, and SuPerFed. These methods represent state-of-the-art approaches that address local adaptation by focusing on personalization and interpolating between global and local models.
> Our results demonstrate that while these methods perform well in general FL scenarios, they exhibit limitations when applied to the extreme heterogeneity and dynamic participation characteristic of financial datasets. Specifically, the integration of Taylor-based linearization and probabilistic adaptation enables precise and computationally efficient local updates, outperforming the included baselines in both average and worst-case scenarios.

---

> ### Comment · Reviewer_fzmC · 2024-12-13
> **Post-rebuttal comment**
>
> Thank you for the author's rebuttal. The author has successfully addressed many of my concerns.

---

### Decision · Action_Editor_NTWv · 2024-12-25

**Recommendation:** Accept with minor revision

**Comment:**

This paper presents an empirical study on federated learning, with an emphasis on financial applications. A new algorithm is proposed which consists of two stages: a global training stage utilizing the data across all nodes to obtain a shared model parameter, and a local adaptation stage that fine-tunes this globally-trained model using local data. In particular, instead of using SGD for fine-tuning, the local adaptation stage linearizes the globally-trained model before applying the quadratic loss, which converts the conventional SGD-based fine-tuning to least square. Experiments are performed to demonstrate the good performance of this approach in a variety of settings.

On the positive side, all reviewers found the paper well-motivated. A notable strength is that the proposed algorithm is tested in a fairly large amount of settings, against plenty of baselines. Reviewers also found the related works to be sufficiently discussed.

However, currently there are two major issues obstructing an unconditioned acceptance decision.
- Despite the variety of tested baselines, an important one is missing: using SGD in the local adaptation stage (initialized at the model parameter obtained from global training). Comparing to this baseline is crucial for justifying the main algorithmic components proposed by this paper, namely linearization and least square. This is different from the Individual Train baseline considered in Section 5.1.1. Admittedly, it's possible that such a SGD-fine-tuning baseline is already considered in this paper but all reviewers and I missed it. In that case, the authors might add a note, or certain pointers in the paper to make it clearer.
- Some theoretical analysis is presented in Appendix A, but after a skimming I found it unclear and sloppy. A requested change is to reorganize this section into a theorem-proof format, with all assumptions clearly stated in or before the theorem. Then, certain components of the proof could be separately presented as lemmas, which will improve its readability. I also encourage the authors to reuse existing theoretical results, instead of proving everything from scratch. Is the analysis following any existing analytical template? If so then this needs to be made clear.
- Rather than revising Appendix A, an alternative option is to remove it altogether. After all this is primarily an empirical paper, and it is fine to present an algorithm with very good empirical performance but no proofs.

Summarizing the above, I would recommend acceptance conditioned on the above two issues being properly addressed.

Other suggestions and minor issues

- The authors are also suggested to polish the writing in the revision. Just to name a few issues:

1. The inline citations should be (author, 2024) rather than author (2024), unless the name of the author is a part of the sentence.

2. Showing the loss values in the abstract has no tangible meaning without introducing the experiment setup, therefore it is suggested to remove those.

3. Another suggestion is to add a pseudocode of the proposed algorithm, and clearly introduce the required hyperparameters.

4. Overall, the authors may also consider being more concise and crisp in the writing style. Currently the paper seems unnecessarily lengthy. Certain parts of the paper, e.g., Section 4.2, are removable.

- On the technical side, I encourage the authors to consider different settings of conditioning, i.e., the eigenvalues of the matrix $JJ^T$. The degenerate case ($JJ^T$ is not invertible) needs to be addressed in the presentation of the algorithm.

To finally summarize, the reviewers and I find more strengths than weaknesses in this paper, therefore we would be glad to support its acceptance to TMLR, as long as the above two major issues are properly addressed in the final revision. Here is a gentle reminder of TMLR's acceptance standard: acceptance is conditioned on the claims being supported by sufficient evidences. To meet this standard, one could either add evidences to support existing claims in a paper, or adjust the claims and clearly state the limitations.

I look forward to checking the final version of this paper.

**Audience:**

Yes, the findings are of interest to the machine learning community.

**Claims And Evidence:**

Partially, yes (regarding whether the claims are sufficiently supported by evidences).

First, extensive experiments are performed to support the claim that the proposed algorithm works well. But if the claim is the empirical *advantage* of the proposed algorithm over existing baselines, then an important baseline is missing (see below for justification).

The theoretical analysis is less satisfactory. Removing it should be fine; otherwise, substantial reorganization is required to make the analysis clear and convincing.

---

> ### Author Response · Authors · 2024-12-31
> **Addressing Missing Baseline and Theoretical Analysis Concerns**
>
> Response to Comment 1 :
>
> Thank you for your follow-up and for highlighting the concerns regarding the direct fine-tuning baseline.
>
> To address this concern, we have implemented and tested the suggested baseline, and the results are included in Section 5.3 of the revised manuscript. This additional comparison provides a more comprehensive evaluation and significantly strengthens the rigor of our analysis, highlighting the clear advantages of FLARE-LA over the baselines. We sincerely appreciate these valuable suggestions, which have helped improve the quality and completeness of our work.
>
> The proposed FLARE-LA consistently outperforms the locally enhanced baselines by effectively leveraging global knowledge while adapting to local requirements. Unlike the baselines, which rely on additional local updates to improve personalization, FLARE-LA incorporates a Taylor-based linearization strategy and a probabilistic adaptation mechanism. These innovations ensure that local updates remain aligned with the global model, avoiding the common drift observed in federated learning settings with heterogeneous data distributions. Such drift often results in a loss of generalization and robustness, particularly when additional local updates dominate the training process. In contrast, FLARE-LA preserves the collaborative training efforts of the FL stage while enhancing local performance in a balanced and systematic manner.
>
> The experimental results confirm that while baselines with extra local updates show minor improvements in average test loss, their risk metrics, e.g., VaR95\% and CVaR95\%, demonstrate significant instability, especially as the number of local updates increases. FLARE-LA consistently maintains superior performance across all metrics, including substantially lower VaR95\% and CVaR95\% values, underscoring its robustness in both generalization and worse-case performance. In comparison, the baselines demonstrate performance variability and degradation in risk metrics.
>
> Simply using the FL-trained global model as an initial point for additional local updates compromises the model's ability to handle information beyond the local datasets. Our method addresses this challenge by maintaining a systematic balance between global and local contributions. This is achieved through the integration of Taylor-based linearization, which enhances the generalization and probabilistic adaptation, ensuring that local updates adjust to specific local needs without sacrificing the applicability of the global model. These carefully designed mechanisms allow FLARE-LA to achieve stability and robustness.
>
> Response to Comment 2 :
>
> We appreciate the feedback regarding the theoretical analysis in Appendix A. Based on the suggestion, we have opted to remove Appendix A from the revised paper. Instead, we have redirected the focus to an enhanced analysis in Section 4.2, which provides a more detailed and rigorous examination of the robust local adaptation mechanism within the FLARE-LA framework.
>
> The revised Section 4.2 adopts a structured and cohesive approach to the analysis, ensuring that all assumptions are explicitly stated and that the results align with existing theoretical foundations. By incorporating a clear and concise explanation, the analysis highlights the theoretical underpinnings of our robust local adaptation strategy without introducing unnecessary complexity. This change enhances the paper's readability while maintaining the scientific rigor necessary to support our empirical findings.
>
> We believe that this adjustment addresses the concerns while aligning the paper's focus with its empirical contributions. Thank you for your constructive input.
>
> Response to Comment 3 :
>
> Thank you for your detailed feedback and valuable suggestions! We have made the following revisions to address the comments:
> 1. We have revised all inline citations as suggested.
> 2. The loss values mentioned in the abstract have been removed.
> 3. The pseudocode of the proposed algorithm has been added to Section 4.1. Additionally, we have introduced the required hyperparameters with detailed explanations of their roles.
> 4. The paper has been revised for improved conciseness and clarity. We have removed the Appendix A, and rewritten Section 4.2 to include essential analysis.
> 5. Section 4.1 has been updated to address different settings of conditioning, including degenerate cases where the matrix is not invertible.